# Increased cortical plasticity leads to memory interference and enhanced hippocampal-cortical interactions

Irene Navarro Lobato[1]*, Adrian Aleman-Zapata[1], Anumita Samanta[1], Milan Bogers[1], Shekhar Narayanan[1], Abdelrahman Rayan[1], Alejandra Alonso[1], Jacqueline van der Meij[1], Mehdi Khamassi[2], Zafar U Khan[3], Lisa Genzel[1]*

[1]Donders Institute for Brain, Cognition and Behaviour, Radboud University Nijmegen, Nijmegen, Netherlands; [2]Institute of Intelligent Systems and Robotics, CNRS, Sorbonne Université, Paris, France; [3]Laboratory of Neurobiology, CIMES, University of Malaga, Malaga, Spain

**Abstract** Our brain is continuously challenged by daily experiences. Thus, how to avoid systematic erasing of previously encoded memories? While it has been proposed that a dual-learning system with 'slow' learning in the cortex and 'fast' learning in the hippocampus could protect previous knowledge from interference, this has never been observed in the living organism. Here, we report that increasing plasticity via the viral-induced overexpression of RGS14414 in the prelimbic cortex leads to better one-trial memory, but that this comes at the price of increased interference in semantic-like memory. Indeed, electrophysiological recordings showed that this manipulation also resulted in shorter NonREM-sleep bouts, smaller delta-waves and decreased neuronal firing rates. In contrast, hippocampal-cortical interactions in form of theta coherence during wake and REM-sleep as well as oscillatory coupling during NonREM-sleep were enhanced. Thus, we provide the first experimental evidence for the long-standing and unproven fundamental idea that high thresholds for plasticity in the cortex protect preexisting memories and modulating these thresholds affects both memory encoding and consolidation mechanisms.

*For correspondence:
ire_nl@hotmail.com (INL);
l.genzel@donders.ru.nl (LG)

**Competing interest:** The authors declare that no competing interests exist.

## Editor's evaluation

This important study reveals that slow plasticity in the neocortex is essential to prevent memory interference. The method of artificially increasing plasticity in the prefrontal cortex of rats during learning and its effect on sleep physiology, when memories are believed to be reprocessed, is solid. The work will be of interest to neuroscientists interested in learning and memory.

## Introduction

Since patient H.M. (*Scoville and Milner, 1957*) we know that memories are supported in the brain by a dual-learning system, but why this is the case remains unclear. Initially memories are stored in the hippocampus via synaptic changes in this more plastic brain area, known as the 'fast learner' (*Marr, 1970*). Later during sleep these hippocampal representations support reactivations of recent memories in the neocortex, the 'slow learner' in the brain. Neocortical synapses are less plastic and therefore are thought to change only a little on each reinstatement. Therefore, remote memory is based on over time accumulated neocortical changes, potentially enacted during post-training consolidation mechanisms during sleep. Computational models testing why we have a dual-learning system have proposed that the neocortex learns slowly to discover the structure in ensembles of experiences

(*Marr, 1970*; *McClelland et al., 1995*; *Marr, 1971*). Further, the hippocampus would then still permit rapid learning of new items without disrupting this structure and therefore the dual system would protect our memories from interference when new memories would overwrite existing ones without the dual system. Although these theories provide remarkable insights about learning and knowledge extraction, they remain computational models with – until now – no direct experimental support, due to the lack of a valid behavioral paradigm that enables examining structured knowledge extraction in rodents as well as interference effects.

To test if naturally restricted plasticity in the neocortex protects from memory interference, we artificially increased plasticity in the prelimbic cortex via the overexpression of an established plasticity-enhancer called regulator of G protein signaling 14 of 414 amino acids (RGS14414) (*Navarro-Lobato et al., 2021*; *Masmudi-Martín et al., 2019*). The overexpression of RGS14414 is known to lead to increased BNDF and dendritic branching in the targeted area (*Navarro-Lobato et al., 2021*; *Masmudi-Martín et al., 2019*) and thereby increase plasticity locally. This increased local plasticity makes memories, that usually would not be retained, last longer and can rescue memory-deficits accompanying aging or diseases (*Navarro-Lobato et al., 2022*; *Masmudi-Martín et al., 2020*; *López-Aranda et al., 2009*). However, until now, the prefrontal cortex had not been targeted and it remained unknown how increasing plasticity would affect previously acquired knowledge. We combined this plasticity manipulation in the prefrontal cortex with a novel behavioral task – the Object Space task (*Genzel et al., 2019*) – that allows the testing of semantic-like as well as simple memories in rodents.

The Object Space task is based on the natural tendency of rodents to explore more novel objects in more novel locations. Most object-exploration tasks use this tendency to test if animals remember the previous trial, in contrast the Object Space task consists of multiple trials – for rats five training trials – with conditions differing in the underlying statistical structure of possible object-locations in these trials. The key condition testing for the behavioral expression of semantic-like memory (Overlapping) has one location that always contains an object, while the second object can be in one of three other locations. Critically, the final training and test trial 24 hr later have the same spatial configuration, thus the animal will only show preferred exploration of the less often presented location, if they created a cumulative memory over multiple trials. The main control condition (Stable) has the same configuration for the five training trials and one new location at test. Thus, this simple-memory, control condition can be solved by either remembering the final trial or multiple trials.

Here, we adapted the Object Space task to be able to contrast how later experiences interfere with the memory of the training trials. For this, the effect of the 24 hr trial, which now functions as interference trial violating previous trained location-rules, on a newly added test trial 48 hr later (72 hr after original training) was investigated. We show that increased cortical plasticity leads to better one-trial memory performance in a simple, one-trial object location task. However, we observed that such enhanced fast learning is associated with impaired semantic-like, cumulative memories tested in the Object Space Task. In alignment with these findings, pharmacological experiments confirmed that these results were an outcome of local changes in the prelimbic cortex. Next, we assessed the learning rate by devising a computational model and this model revealed that an increased learning rate in the intervention group augmented the influence of recent in contrast to remote memories on behavior. Finally, electrophysiological experiments showed that increased plasticity leads to (1) less NonREM sleep and smaller delta waves, (2) more neurons with slower firing rates, (3) increased hippocampal-cortical connectivity measured in theta-coherence, delta-spindle-ripple coupling and increased granger causality during ripple events and (4) off-target changes in hippocampal ripples.

## Results
### Increasing cortical plasticity leads to more memory interference

Prelimbic plasticity was increased by the overexpression of RGS14414 (*Navarro-Lobato et al., 2021*; *Masmudi-Martín et al., 2019*; *Navarro-Lobato et al., 2022*; *Masmudi-Martín et al., 2020*; *López-Aranda et al., 2009*; *Figure 1A and B*) and initially two behavioral experiments were performed using the Object Space task (*Genzel et al., 2019*). In this task, the condition Overlapping tests for the extraction of an underlying structure across five training trials, while the Stable condition tests for the simple memory of the last experience (*Figure 1C*). In both experiments, we examined the effect of an interference trial 24 hr after initial training, with object configurations violating previously trained

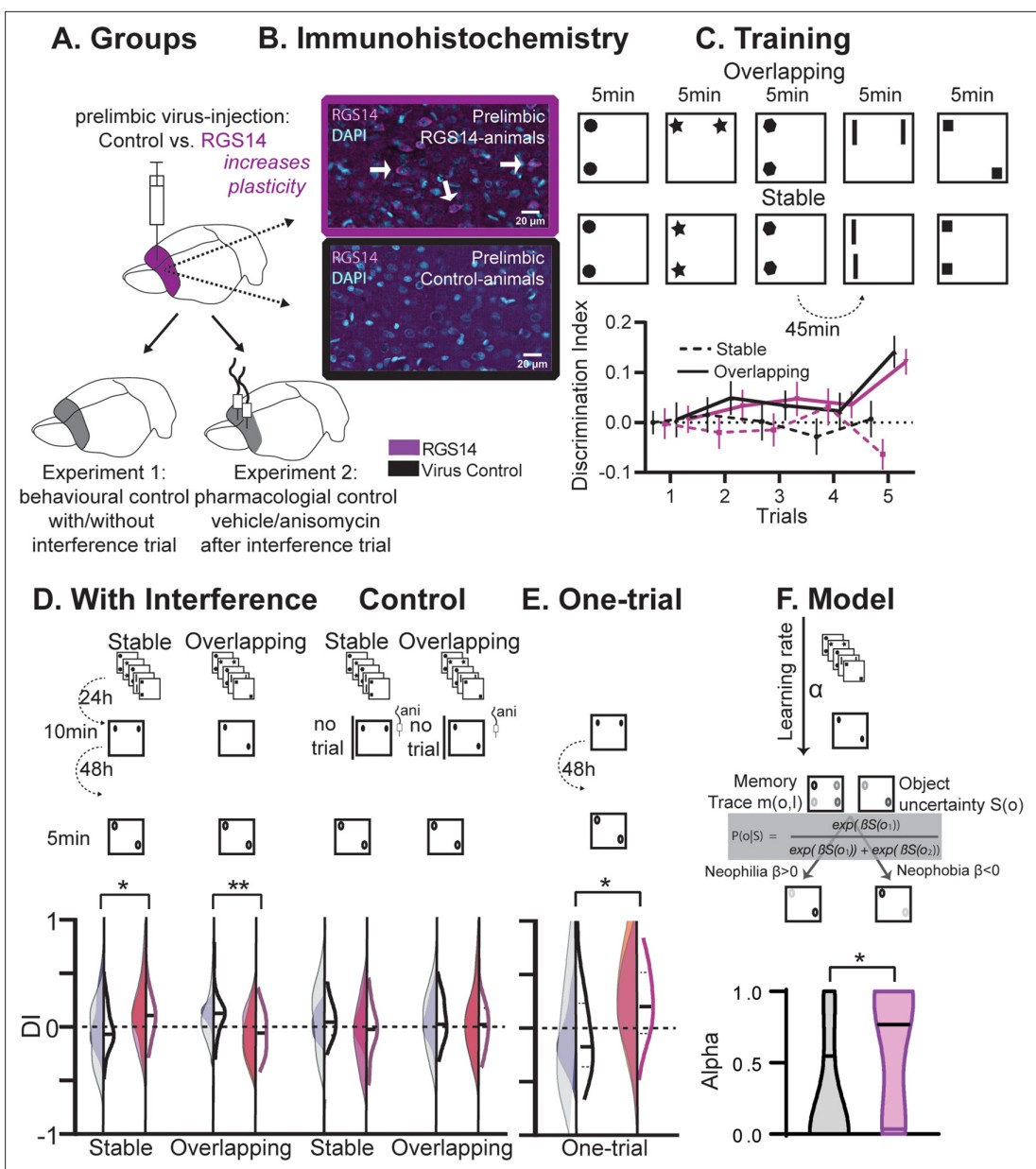

**Figure 1.** Behavioral experiments. (**A**) Half the animals were injected with a lentivirus for the overexpression of RGS14414 increasing plasticity in the prelimbic cortex, the other half had a control virus. These animals were included in either Experiment 1 (behavioral control), or were implanted with canula to the prelimbic cortex for Experiment 2 (pharmacological controls, total n=65 with n=16–17 per experimental group). (**B**) Immunohistochemistry for RGS14 expression in treated and control animals (purple, cyan DAPI staining). (**C**) Object Space Task training contains five trials with 45 min inter-trial-intervals. In Overlapping one location always contains an object, while the other object moves each trial. In Stable the configuration always remains the same. Discrimination Index (exploration time moved-not moved/sum) over training trials show slowly rising discrimination in Overlapping and not Stable with preference for the less often used locations especially in the 5th trial as expected (*Genzel et al., 2019*) (rmANOVA condition $F_{1,65}=16.9$ p<0.001, conditionXtrial $F_{4,260}=3.3$ p=0.012, all other p>0.37, DI interference and exploration times see *Figure 1—figure supplement 1*). Of note, object locations and configurations were counterbalanced across animals. (**D**) 24 hr after training animals had an interference trial (10 min, different and same configuration as last training trial for Stable and Overlapping, respectively) followed by a test trial another 48 hr later (again different and same configuration as interference trial for Stable and Overlapping, respectively). In the control conditions (right side), for Experiment 1 there was no interference and in Experiment 2 animals were infused with anisomycin after the interference trial. Virus control grey, RGS-overexpressing purple, lighter shades experiment 1 (with or without interference trial), darker shades experiment 2 (with or without anisomycin infusion). There was a significant interaction where interference had the opposite effect in each condition according to virus manipulation (rmANOVA with condition, interference/drug, experiment, virus; cond*int/drug*virus $F_{1,61}=13.2$ p<0.001; stable with interference $t_{63}=2.1$ p=0.039, overlapping with interference $t_{63}=3.1$ p=0.003, other p>0.12). (**E**) One-trial control followed 48 hr later by test, RGS14-overexpressing animals performed better than controls ($t_{30}=2.2$ p=0.037). (**F**) Model-fitting show that RGS14-overexpressing animals have a higher learning rate α (KS-D=0.46 p=0.044). *p<0.05, **p<0.01.

*Figure 1 continued on next page*

*Figure 1 continued*

The online version of this article includes the following figure supplement(s) for figure 1:

**Figure supplement 1.** Object Space Task.

rules, on a test trial 48 hr later (*Figure 1D*). The design was such that in the Overlapping condition remembering the training resulted in positive discrimination indices at test, in contrast the interference would lead to 0. In the control condition Stable remembering both the training or the interference would result in positive discrimination indices. Performance on the interference trial (*Figure 1—figure supplement 1*) tests the memory for the original training trials (*Genzel et al., 2019*). Both groups performed above chance for both conditions, evidence that later differences were not due to differences in the original memory but instead effects of the interference trial.

At the test trial conducted after interference, RGS14-overexpressing animals exhibited higher discrimination indices in the Stable condition (*Figure 1D*, $p < 0.05$) but lower in the Overlapping condition ($p < 0.01$) in comparison to controls. The results in the Overlapping condition emphasize that in controls cumulative memory expression was protected from interference, but after increasing plasticity in the cortex interference effects were observed. Further, the simple memory in the Stable condition did not last until test in controls, however after increasing plasticity the memory lasted longer.

To show that these effects were a result of the interference, we conducted three control experiments. Firstly, we performed a behavioral control, in which animals did not experience the interference trial. Secondly, we performed a pharmacological control, in which animals did experience the interference, but any subsequent plasticity-related changes were inhibited in the cortex via the infusion of anisomycin, a protein synthesis inhibitor. In these experiments RGS14-overexpressing animals showed discrimination indices comparable to controls emphasizing the determinant role of the interference trial in producing opposite outcomes. Thirdly, we added an additional control, in which animals did not receive pretraining on the Object Space Task conditions, but instead were only exposed to an object configuration for one-trial. When tested 48 hr later, increased cortical plasticity led to an enhanced one-shot memory performance (*Figure 1E*, $p < 0.05$).

Together, the behavioral and pharmacological results show that increasing cortical plasticity with RGS14-overexpression caused larger interference effects in a semantic-like memory task. Increased interference in this case was due to better memory for one-trial experiences and local changes in the prelimbic cortex. Thus, our experimental results verify for the first time the hypothesis that lower cortical plasticity is critical to protect previous knowledge from interference effects.

## Memory interference is due to a higher learning rate

To characterize the build-up of a memory trace and its expression in the Object Space Task, we previously developed (*Genzel et al., 2019*) a computational model that progressively learns place–object associations and makes decisions about which proportion of time to spend exploring each object in order to minimize uncertainty about these place–object associations. The model employs two parameters: a learning rate $\alpha$, which determines the balance between recent and remote memories, and a parameter $\beta$, which determines the balance between neophilic (preference for more novel object location) and neophobic (aversion for more novel object location) exploratory behaviors. Here, we fitted our model on the behavioral data-set to find for each individual subject the values of the model parameter set $\alpha$ and $\beta$ that best fit the data. There was no difference in memory expression ($\beta$). However, RGS14-overexpressing animals had systematically higher learning rate ($\alpha$) values (*Figure 1F*, $p < 0.05$). This indicates that exploration behavior in RGS14-overexpressing animals was driven more by recent than remote memories in contrast to controls.

Thus, the modeling results show that increasing cortical plasticity with RGS14-overexpression caused larger interference effects in a semantic-like memory task due to a higher learning rate.

## Increased cortical plasticity results in shorter NonREM bouts and smaller delta-waves

Sleep is supposedly the price the brain pays for plasticity (*Tononi and Cirelli, 2014*; *Tononi and Cirelli, 2006*). The idea is that during a waking episode, learning statistical regularities about the current environment requires strengthening connections throughout the brain. This increases cellular

needs for energy and supplies, decreases signal-to-noise ratios, and saturates learning. Therefore, subsequently during sleep, previous waking activity would lead locally to larger delta-waves (1–4 Hz) and spontaneous activity during these oscillations in NonREM sleep should renormalize net synaptic strength and restore cellular homeostasis (*Huber et al., 2004*).

To test this, after viral-injection rats were implanted with hyperdrives containing 16 tetrodes targeting the hippocampus and prelimbic cortex (*Figure 2*). We recorded each day 7 hr of neural activity during training as well as sleep in the Object Space task (OS) and compared this to a home cage control (HC). Surprisingly, RGS14-overexpressing animals showed less NonREM sleep (p<0.05, *Figure 2C* combining all sleep periods in the 7 hr recording day), which can be attributed to shorter bout lengths (p<0.0001, *Figure 2D*). They also had slightly longer REM bout lengths (p=0.048, sleep stages over time *Figure 2—figure supplement 1*).

NonREM sleep is dominated by alternations of on and off-periods of neural activity that can be detected in the LFP signal as delta-waves. We detected delta-waves in the cortex and hippocampus, surprisingly in both groups the hippocampus presented with a higher rate of delta-oscillations (p<0.0001, *Figure 2E*). In addition, RGS14 showed even higher rates than controls in the hippocampus (p<0.0001). But only in controls, we observed that cortical delta-waves occurred more after learning (OS vs HC, p<0.0001, *Figure 2E*) while their intrinsic frequency decreased (*Figure 2—figure supplement 2*). Interestingly, delta amplitudes presented with a different pattern. In controls, delta-waves were smaller in the hippocampus in contrast to the cortex (p<0.0001), but in RGS14-overexpressing animals cortical delta-waves were as small as their respective hippocampal delta-waves and significantly smaller than cortical delta-waves in controls (p<0.0001). In both groups and brain areas delta amplitude increased after learning (all OS vs HC p<0.0001).

Combining this analysis with analysis of neuronal activity (spikes) lets us determine more accurately the length of the respective on and off periods underlying the delta wave. RGS14-overexpressing animals presented with longer off periods than controls (p<0.01). However, known homeostatic changes, where off-period durations decrease over longer sleep periods, remained the same in both groups (Post Trial period X type interaction p<0.0001 but no interaction with group all p>0.8).

In sum, in controls we could confirm the proposition that learning and therefore plastic changes lead to a homeostatic response with increases in delta wave activity. However, we show that artificially increasing plasticity in the prelimbic cortex does not lead to a simple enhancement of this effect. Instead, we observed in RGS14-overexpressing animals that cortical delta-waves become as small as hippocampal delta-waves, off-periods become longer and less NonREM sleep is obtained.

## More neurons with slower firing rates after plasticity-increase

Next, we focused on neuronal firing of individual neurons in the prelimbic cortex. We first determined the firing rate of each neuron during task performance and noticed that RGS14-overexpressing animals showed more neurons with low firing rates (p<0.0001, *Figure 2H*). When splitting neurons in controls into pentiles according to their firing rate and applying the same margins to RGS14-overexpressing animals, the two lowest firing rate groups represented 40% of the neurons in controls but 75% in RGS14 (p=0.0005, *Figure 2I*, same for split performed on wake in sleep box or sleep states see *Figure 2—figure supplement 2*).

Moving to neuronal firing during sleep, only faster-firing neurons showed decreases in firing rate across different wake and sleep states (*Figure 2—figure supplement 2*) and this was the same for both groups. Spikes were less phase-locked to the slow oscillations in RGS14-overexpressing animals, which was seen for all firing rate groups (*Figure 2J*, p<0.0001). However, faster-firing neurons contributed the most spiking activity to the upstate and with a decrease of these neurons in RGS14, these animals presented less spikes in the upstate (*Figure 2J*). This decrease in upstate activity likely underlies the decrease in delta-wave amplitude.

To ensure firing rate differences were not a result of differences in learning itself, firing rates during the period before the first trial was calculated. Most animals did not sleep in this period, thus we focused on wake firing rates and the same difference was present (*Figure 2K*, p<0.0001). Next, firing rates during the OS trials as well as during the periods directly after the trials (PT 1–8 wake and NREM) were normalized to pre-task rates (*Figure 2L*). In RGS14 there was a larger increase in firing rates from pre-task in comparison to controls (p<0.0001), further evidence that in RGS14 neurons were more plastic and changed more due to experience. Interestingly, RGS14 generally increased their

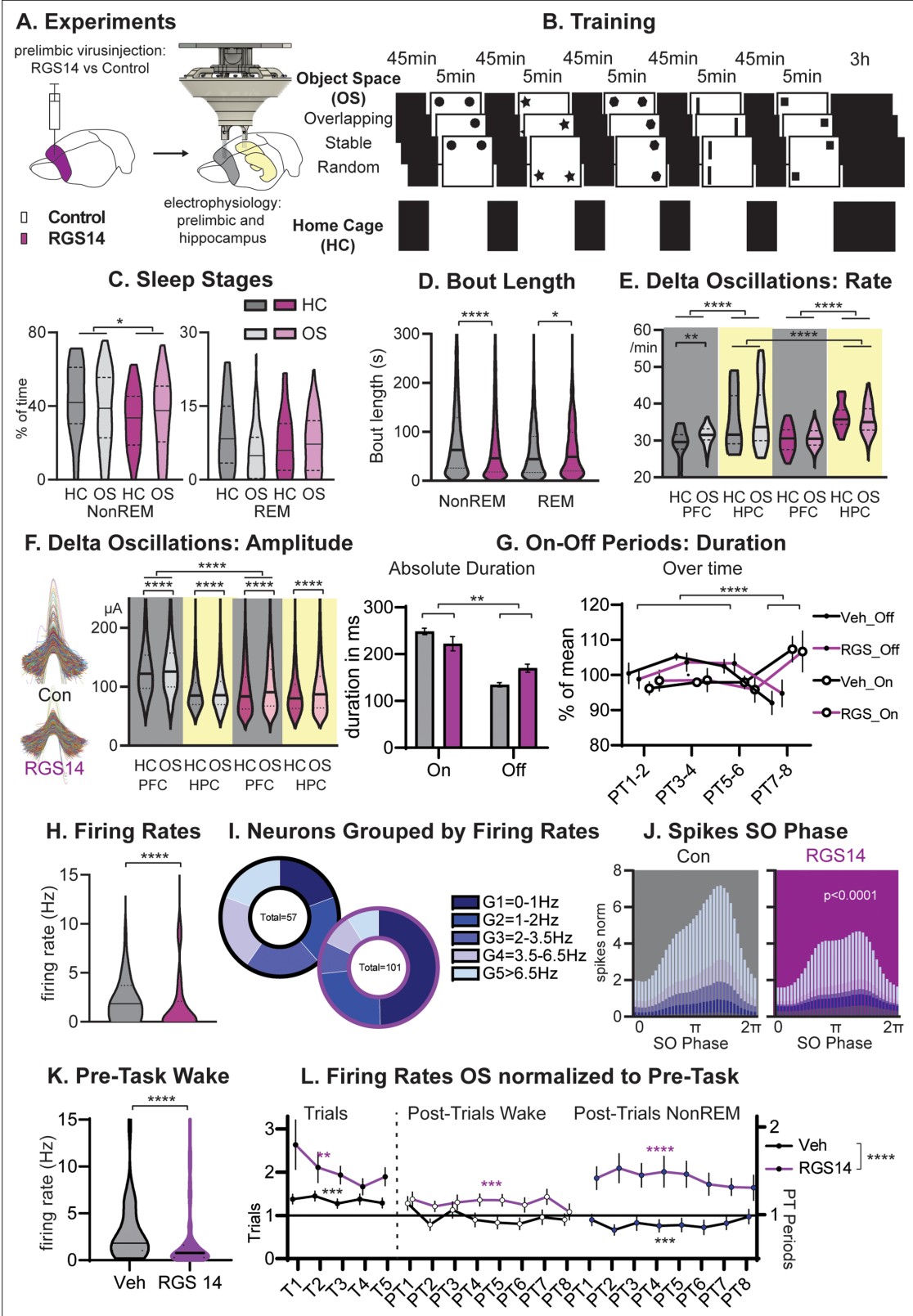

**Figure 2.** Sleep and Firing Rates. (**A**) Animals received RGS14 overexpressing (n=4) or control (n=4) virus and were implanted with a hyperdrive (10 tetrode prelimbic, 6 tetrodes hippocampus) three weeks later. (**B**) Animals ran the three conditions of the Object Space Task (OS, Stable and Overlapping as described above, Random with constantly moving objects) and a home cage control (HC, same structure of the day but remained awake in recording box during trial periods). (**C**) Controls had more NonREM sleep (KS-D=0.17 p=0.05) and no change in REM sleep (KS-D=0.15

*Figure 2 continued on next page*

*Figure 2 continued*

p=0.12). (**D**) NonREM bout length was longer in controls (MW U=6887019 p<0.0001) but REM length was shorter (MW U=501,849 p=0.048. (**E**) In both groups, hippocampal delta rates were higher than cortical rates (p<0.0001). The rate of delta oscillations in the cortex increased in controls (KS-D=0.11 p=0.007) but not in RGS14 after OS, the rate of hippocampal delta oscillations was overall higher for RGS14 than controls (KS-D=0.31; p<0.0001). (**F**). In controls delta oscillations were larger in the cortex than hippocampus, in contrast in RGS14 the cortex showed the same amplitude delta-waves as the hippocampus (Cortex Con vs RGS14 KS-D=0.34 p<0.0001). In both groups and brain areas delta amplitude increased after OS (KS-D >0.02 p<0.0001)). (**G**) There was a significant interaction in duration of On-Off periods underlying delta-waves with RGS14-overexpressing animals presenting with longer Off-periods (groupXtype interaction ANOVA $F_{1,164}$=9.0 p=0.0031, type $F_{1,164}$=65.0 p<0.0001). But the time course of on-off durations did not differ between groups (PTXtype interaction ANOVA $F_{3,152}$=946.6 p<0.0001, no interaction with group all p>0.8) (**H**) RGS14-overexpresssing animals had more neurons with lower firing rates (KS-D=0.36 p<0.0001, for split by other states see *Figure 2—figure supplement 2*). (**I**) Division of neurons according to their firing rates (Chi-square$_4$=20.13 p=0.0005). (**J**) Spikes were less phase locked to the slow oscillation phase (circ stats p<0.0001 for each neuron group) and less G4-5 neurons led to less spikes during the upstate in RGS14. (**K**) Firing rates during the pre-task period, only including wake since few animals slept (KS-D=0.41 p<0.0001) (**L**) Firing rates on OS days normalized to Pre-Task (wake). From left to right during task (trials, left), post-trial periods wake in recording box, and during NonREM sleep. RGS14 overall showed larger increases in firing rates from pre-task than controls (ANOVA virus $F_{1, 2545}$ = 52,43 p<0.0001). In all time periods RGS14 showed an increase to pre-task (one-sample t-test to 1 P=0.003–0001), in Controls during task there was an increase (p=0.0004) during PT wake no change (p=0.3), and during PT NonREM a decrease in firing rates (p=0.0005) Control grey, RGS-overexpressing purple, darker shades home cage (HC), lighter shades Object Space Task (OS), *p<0.05, **p<0.01,****p<0.0001.

The online version of this article includes the following figure supplement(s) for figure 2:

**Figure supplement 1.** Wake and sleep states for each 5 min bin the sleep recording box.

**Figure supplement 2.** Oscillations and Firing Rates.

**Figure supplement 3.** Placement of tetrodes per rat and brain area.

firing rates over the day and presented with higher rates in comparison to pre-task during all time periods (one-sample t-test to 1 p=0.003–0001). In contrast, for Veh during task there was an increase (p=0.0004) but then during following NonREM a decrease in firing rates (p=0.0005) in comparison to pre-task. There was no change for PT wake. These effects were not seen on home cage days (*Figure 2—figure supplement 2*).

To summarize, RGS-overexpressing animals had more prelimbic neurons with slower firing rates. These results provide the first causal evidence that increasing synaptic plasticity shifts the neural firing towards the slow firing end of the neural firing spectrum. Furthermore, because it is the faster-firing neurons that dominate upstate spiking activity and therefore delta amplitude, the slowing of firing rates in the more plastic neurons is most likely the cause of the smaller delta waves seen in these animals.

## Increased hippocampal-cortical connectivity during wake and sleep

Interactions between the hippocampus and cortex are critical during encoding as well as consolidation of memories (*Buzsáki, 1989*; *Figure 3A*). During wake as well as REM sleep these interactions take place in the theta domain and can be measured in theta coherence (*Benchenane et al., 2010*). In NonREM sleep, they can be captured in the coupling of cortical delta and spindle oscillations with hippocampal ripples (*Genzel et al., 2014*). Different types of interactions between these three oscillations have been reported; interactions between two oscillations such as delta followed by spindle (*Mölle et al., 2002*), delta followed by ripple (*Peyrache et al., 2009*), ripple followed by delta (*Maingret et al., 2016*), and spindles with a ripple in their troughs (*Sirota et al., 2003*), but also three-oscillation interactions such as delta followed by spindle with a ripple in the trough (*Diekelmann and Born, 2010*), delta followed by ripple then spindle (*Genzel et al., 2014*), ripple followed by delta and then spindle (*Maingret et al., 2016*). Interestingly, RGS14-overexpressing animals presented with higher hippocampal-cortical theta coherence during both task and REM sleep (*Figure 3B* p<0.001) as well as increased occurrences of all types of NonREM oscillatory coupling (*Figure 3C* p<0.001). In controls, we observed an experience-dependent increase (HC vs OS) in theta coherence (p<0.05) as well as NonREM oscillatory coupling rate (p<0.001, also single spindle rate *Figure 2—figure supplement 2*), confirming the proposed association of these events to learning. This experience-dependent change was absent in RGS14 and higher values were already present in home cage.

Therefore, we could show that increased cortical plasticity led to increased cross-brain interactions during wake and sleep, emphasizing that decreased interactions and decoupling of the cortex from the hippocampus enables the protection of older, cortical memory representations. Further,

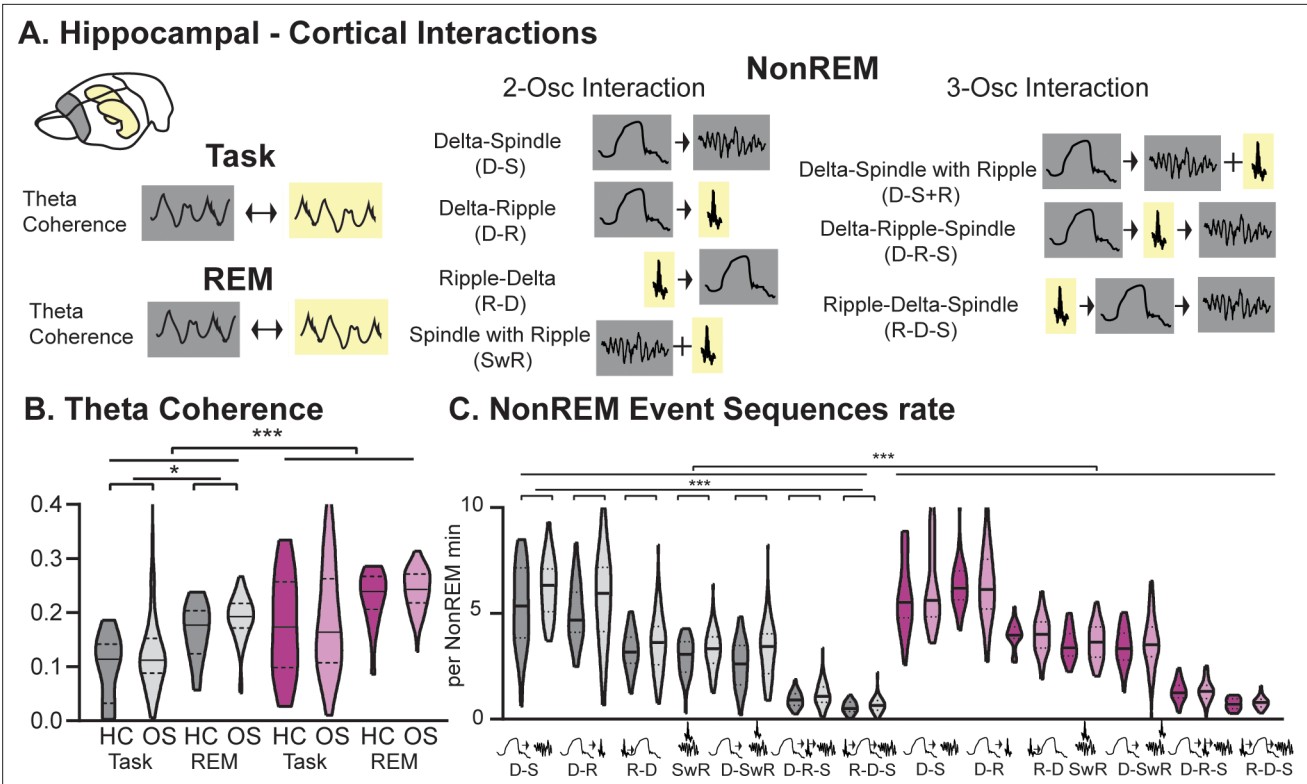

**Figure 3.** Hippocampal-Cortical interactions. (**A**) Brain area interactions are known in Task and REM as theta coherence (*Benchenane et al., 2010*), in NonREM as oscillatory coupling. (**B**) RGS14 showed higher theta coherence (treatment $F_{1,285}$=31.8 p<0.001). Only controls showed higher coherence after learning (controls OS vs HC $F_{1,144}$=5.2 p=0.024, RGS14 OS vs HC $F_{1,141}$=0.6 p=0.44). (**C**) RGS14 showed generally higher rates of oscillations sequences (treatment $F_{1,856}$=18.7 p<0.001), but only in controls could a learning-dependent increase be observed (controls OS vs HC $F_{1,420}$=12.3 p=0.001, RGS OS vs HC $F_{1,436}$=1.3 p=0.25). Control grey, RGS-overexpressing purple, darker shades home cage (HC), lighter shades Object Space Task (OS) *p<0.05, **p<0.01, ***p<0.001, ****p<0.0001.

these findings also highlight that hippocampal-cortical interactions can be regulated top-down by the cortex and not only by other neuromodulating brain areas or the hippocampus as previously assumed.

## Increasing cortical plasticity leads to changes in hippocampal ripples

Next, we focussed on the hippocampal ripple oscillation in NonREM sleep, which is linked to memory reactivation, and therefore is suggested to be a key player in the memory consolidation process (*Buzsáki, 2015*; *Girardeau et al., 2009*; *Grosmark and Buzsáki, 2016*). Increased cortical plasticity led to more, larger and slower ripples (*Figure 4A* all p<0.0001). Further, learning in comparison to home cage led to a decrease in ripple amplitude and increase in frequency in both animals' groups (each p<0.0001). In RGS14 ripples were less phase-locked to the slow oscillation (p<0.0001) and cortical delta power around ripples was decreased in comparison to controls (*Figure 4B*). Surprisingly, granger values around ripples measuring directional connectivity were higher in RGS14-overexpressing animals for Prl→Hpc in the delta frequency range (1–4 Hz) and in the theta/beta range (5–20 Hz) for Hpc→Prl (*Figure 4B*). The excitatory output of hippocampal ripples can lead to a neuronal response in the cortex (*Wierzynski et al., 2009*). This response was larger in controls in comparison to RGS14 (*Figure 4C* p<0.001) because the average activity of ripple responsive neurons was higher (p<0.05). Further, when split into firing-rate groups, it became noticeable that slow firing-neurons (G1) showed no ripple-responses (*Figure 4D*).

In sum, increased cortical plasticity also influenced the hippocampal ripple oscillation, ripples became larger, more numerous and the corresponding information flow and cortical neural response was attenuated. Our results imply that there is an important top-down, cortical influence on this oscillation beyond the known local hippocampal mechanisms.

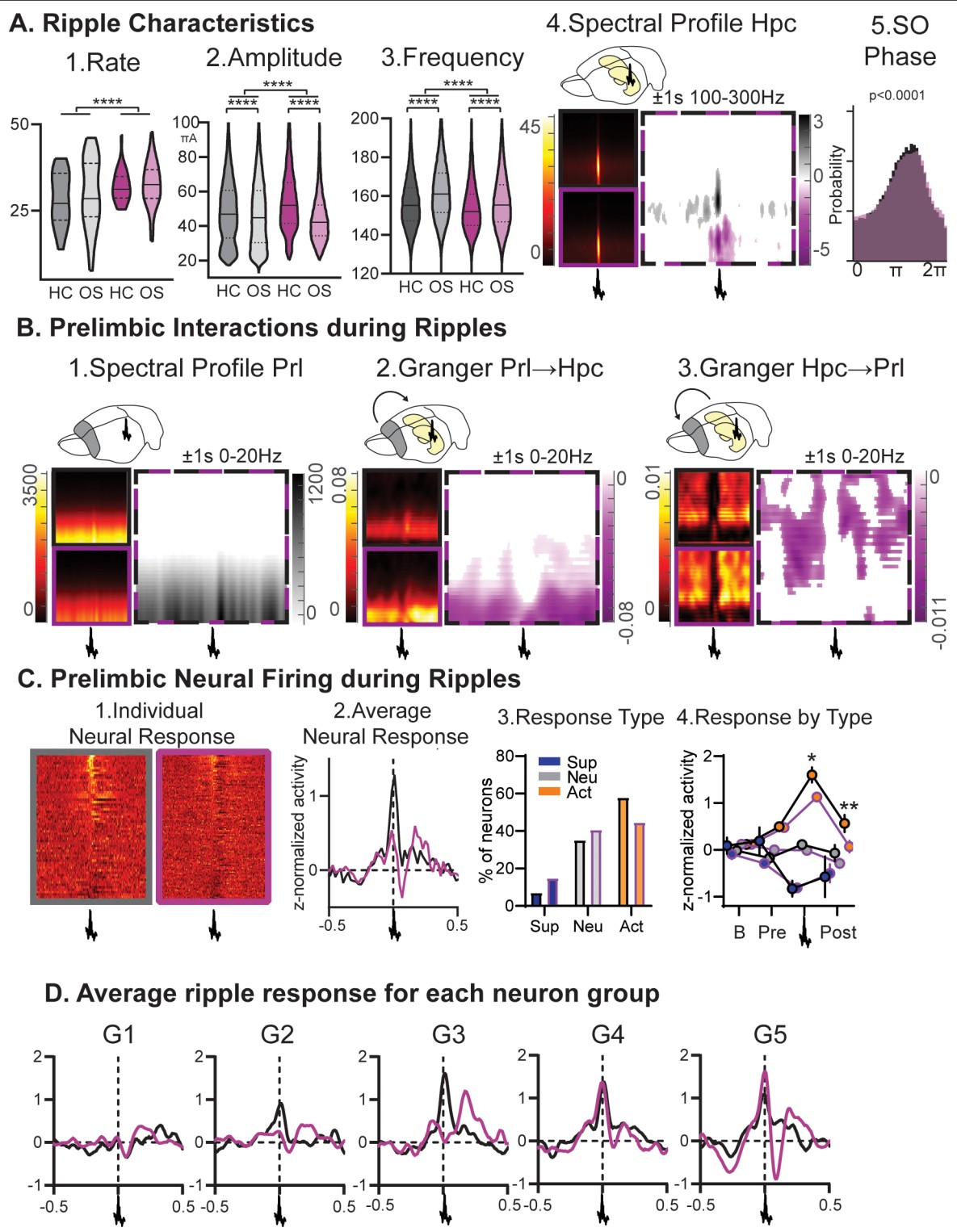

**Figure 4.** Ripples. (**A**) RGS14-overexpressing animals showed changes in 1. ripple rate, 2. amplitude, and 3. frequency (rate KS-D=0.33 p<0.0001, amplitude KS-D=0.13 p<0.0001, frequency KS-D=0.13 p<0.0001). In both groups OS led to a decrease in amplitude and increase in frequency (vehicle KS-D=0.06 p<0.0001, RGS KS-D=0.25 p<0.0001). 4. On the right hippocampal (Hpc) spectral profile 1 s before and after the ripple in the 100–300 Hz range (black Con, purple RGS, dotted Con vs RGS with statistically significant contrast with pixel-based correction for multiple comparison, grey con higher, purple RGS14 higher). 5. Ripples showed less slow oscillation (SO) phase locking in RGS (phase lock circ stat. p<0.0001) (**B**) 1. This was also reflected in decreased delta power around the ripple (Prelimbic PrL spectral profile 1 s before and after the ripple in the 0–20 Hz range). 2. Time-

*Figure 4 continued on next page*

*Figure 4 continued*

frequency granger analysis showed higher delta Prl→Hpc and 3. higher theta/beta Hpc→Prl directional connectivity (black Con, purple RGS, dotted Con vs RGS with statistically significant contrast with pixel-based correction for multiple comparison). (**C**) Neural firing during ripples in prelimbic cortex. From left to right: 1. individual neuron response (each row one neuron) aligned to ripple (middle); 2. average response across neurons (rmANOVA treat*time $F_{98,15288}$=2.7 p<0.001); 3. neurons were categorized into types with ripple suppressed (blue), ripple neutral (grey) and ripple active (orange, Chi-square 3.46 p=0.18); 4.for each types the response at baseline (200–120ms before ripple), pre-ripple (120 ms-40ms before ripple), during ripple (40ms before – 40ms after ripple peak), and post-ripple (40 ms-120ms after ripple, rmANOVA treat*ripple response $F_{2,162}$=3.2 p=0.043, post-hoc RGS vs. Con active neurons during p=0.015 and after p=0.003). (**D**) Same ripple response as in C.2. but now split for the five firing-rate defined neuron groups. Especially faster firing neurons (veh G2-G5, RGS14 G4 and G5) showed higher ripple responses. Control grey, RGS-overexpressing purple, darker shades home cage (HC), lighter shades Object Space Task (OS), *p<0.05, **p<0.01, ***p<0.001, ****p<0.0001.

The online version of this article includes the following figure supplement(s) for figure 4:

**Figure supplement 1.** Classification into ripple-supressed (blue), ripple-neutral (grey) and ripple-active (orange) neurons per firing rate group.

**Figure supplement 2.** Hippocampal (top) and prelimbic (bottom) spectral profile around ripple events for control animals (black) and RGS14 (purple) and the contrast (pixel-based correction for multiple comparison).

**Figure supplement 3.** Hippocampal to prelimbic (top) and prelimbic to hippocampus (bottom) granger time frequency analysis around ripple events for control animals (black) and RGS14 (purple) and the contrast (pixel-based correction for multiple comparison).

## Condition-specific effects

Next, we investigated the different Object Space Task conditions and focused on mechanisms previously proposed by the active systems consolidation theory (*Diekelmann and Born, 2010*; *Genzel et al., 2014*) to be key in the consolidation process: cortical response to ripples and oscillation interactions.

In controls Home Cage presented with less ripple-active neurons than the other OS conditions (*Figure 5*). Interestingly, Overlapping and Random, both conditions in which object-locations change between trials, had more ripple-modulated neurons than the simple-memory condition Stable (p=0.0007). In contrast, after increasing cortical plasticity in RGS14 Home Cage and Stable had the same number of ripple-modulated neurons as the other conditions.

Analyzing the ripple responses revealed that especially activity before the ripple (120–40ms before the peak) differed between conditions and treatments. In controls Overlapping showed the highest activity but all OS conditions differed from home cage (all p<0.05), this was not the case for RGS14.

Next, we examined the fraction of ripples occurring alone or coupled to delta or spindles. In controls, more ripples occurred alone in both the Stable and Home Cage in contrast to Overlapping and Random (p<0.01). However, Overlapping and Random had more ripples following delta waves (large down states, p<0.001). In contrast, in RGS14 Stable and Home Cage had as many ripples following delta as the other conditions, significantly more than the same conditions in controls (p<0.001).

In sum, in controls both Overlapping and Random, conditions with moving locations in the different training trials, had more ripple-modulated neurons and more ripples followed delta-waves than in Stable (simple memory) or Home Cage. In contrast, in RGS14 the same effects were already present in Stable and Home Cage, emphasizing that with increased cortical plasticity all experiences lead to signatures of sleep-related consolidation.

## Discussion

By increasing plasticity in the prelimbic cortex via the overexpression of RGS14 and combining this with behavioral, pharmacological, computational, and electrophysiological approaches, we tested the long-standing but unproven idea of a slow- and fast-learner in the brain. Our data strongly suggests that restricted cortical plasticity is needed to protect memories from interference, the first experimental confirmation of Marr and McClellands ideas that have fundamentally shaped our concepts of memories. Increasing plasticity enhanced one-trial memory, increased memory interference in our semantic-like memory paradigm and learning-rate in our modeling approach. Signatures of cross-brain connectivity – hippocampal-cortical theta coherence, NonREM oscillation coupling, granger causality analysis of ripple events – were all increased after our plasticity manipulation. However, experience-dependent increases, that were present in controls, were lacking in RGS14 animals, because they were already seen in the home cage condition. Increased plasticity in the prelimbic cortex also changed

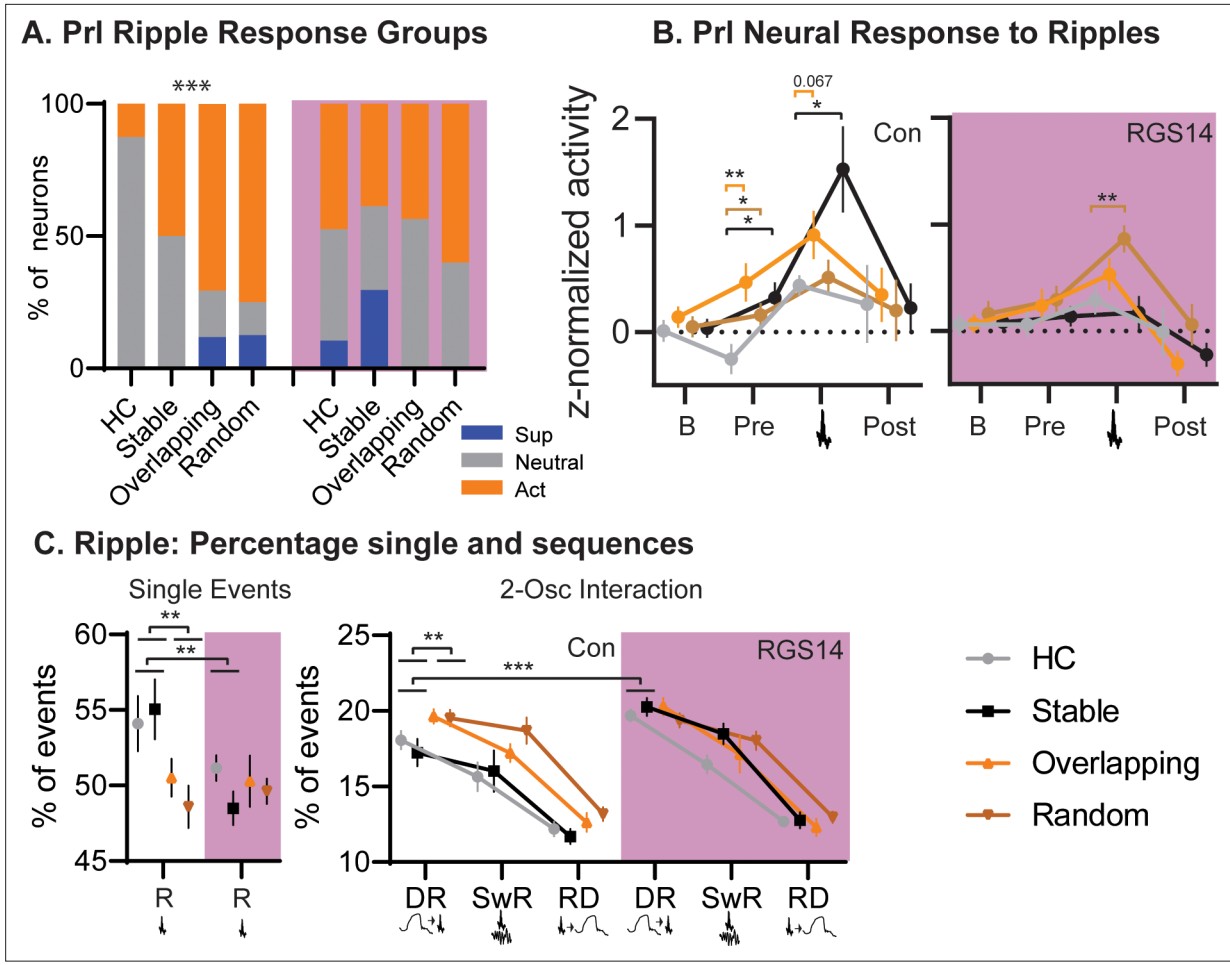

**Figure 5.** Conditions. (**A**) For each condition the percent of neurons that are ripple modulated (suppressed/blue or active/orange) or not (neutral/grey). In controls Overlapping and Random had more ripple-modulated cells, in RGS14 this was already the case for Home Cage and Stable (Chi-square combining Sup and Act to avoid 0, Con Chi-square$_3$=17.1 p=0.0007. RGS14 Chi-square$_3$=3.8 p=0.28.) (**B**). Ripple response for each condition. There was a significant treatment, condition, time interaction (rmANOVA, Time F$_{3,450}$=24.6 p<0.001, treatment F$_{1,150}$=4.4 p=0.037, TimeXtreatment F$_{3,450}$=4.3 p=0.005, TreatmentXCondition F$_{3,150}$=2.9 p=0.038, timeXtreatmentXcondition F$_{9,450}$=1.1 p=0.005). Baseline (200–120ms before ripple), pre-ripple (120 ms-40ms before ripple), during ripple (40ms before – 40ms after ripple peak), and post-ripple (40 ms-120ms after ripple).(**C**). Single and 2-Osc ripple events split for conditions (normalized to count of all ripple events). In vehicles overlapping and random less ripples occurred as single ripples (R) but more followed delta waves (DR), which was not the case for RGS14 (rmANOVA type F$_{3,681}$=2269.9 p<0.0001, TypeXTreatment F$_{3,681}$=4.2 p=0.006, TypeXCondition F$_{3,681}$=2.4 p=0.013, TypeXTreatmentXCondtion F$_{9,681}$=2.3 p=0.018). *p<0.05, **p<0.01, ***p<0.001.

hippocampal ripple oscillations, arguing for a top-down control of the cortex on this oscillation. In controls, we observed that complex learning (Overlapping and Random conditions) but not home cage or simple learning (Stable) led to increased neural response to ripples as well as more ripples that followed delta-waves. In RGS14, these effects were present in all conditions, emphasising that after increasing plasticity even insignificant experiences induce consolidation markers and are retained. Finally, increasing plasticity led to a decrease in firing-rates and smaller delta waves in the cortex, in both cases effects that made the cortical signal more hippocampal-like.

## Effects of different types of learning

The Object Space Task, applied here, allows to separate different types of learning. First off, the simple memory condition Stable, which has the different objects placed at the same two locations throughout training and at test one new location is used, is similar to classic reference memory such as the watermaze (*Samanta et al., 2021*; *Genzel et al., 2017*). Since the different trials have the same configurations, the animal can use both a cumulative memory across trials as well as a recency memory of the last trial to still perform above chance at test. In contrast, in the Overlapping and

Random conditions the objects will be presented in different locations from trial to trial, this trial-by-trial novelty should encourage the animal to consolidate these experiences during sleep and abstract the underlying cumulative statistical distributions. In Overlapping this distribution is skewed, such that one location will always contain an object. Thus, we can test for the behavioral expression of the cumulative memory, by presenting the same spatial configuration at test as it was presented during the final trial. Now, only if the animal extracted the cumulative statistics during all training trials, will there be a preference for the less often shown location at test.

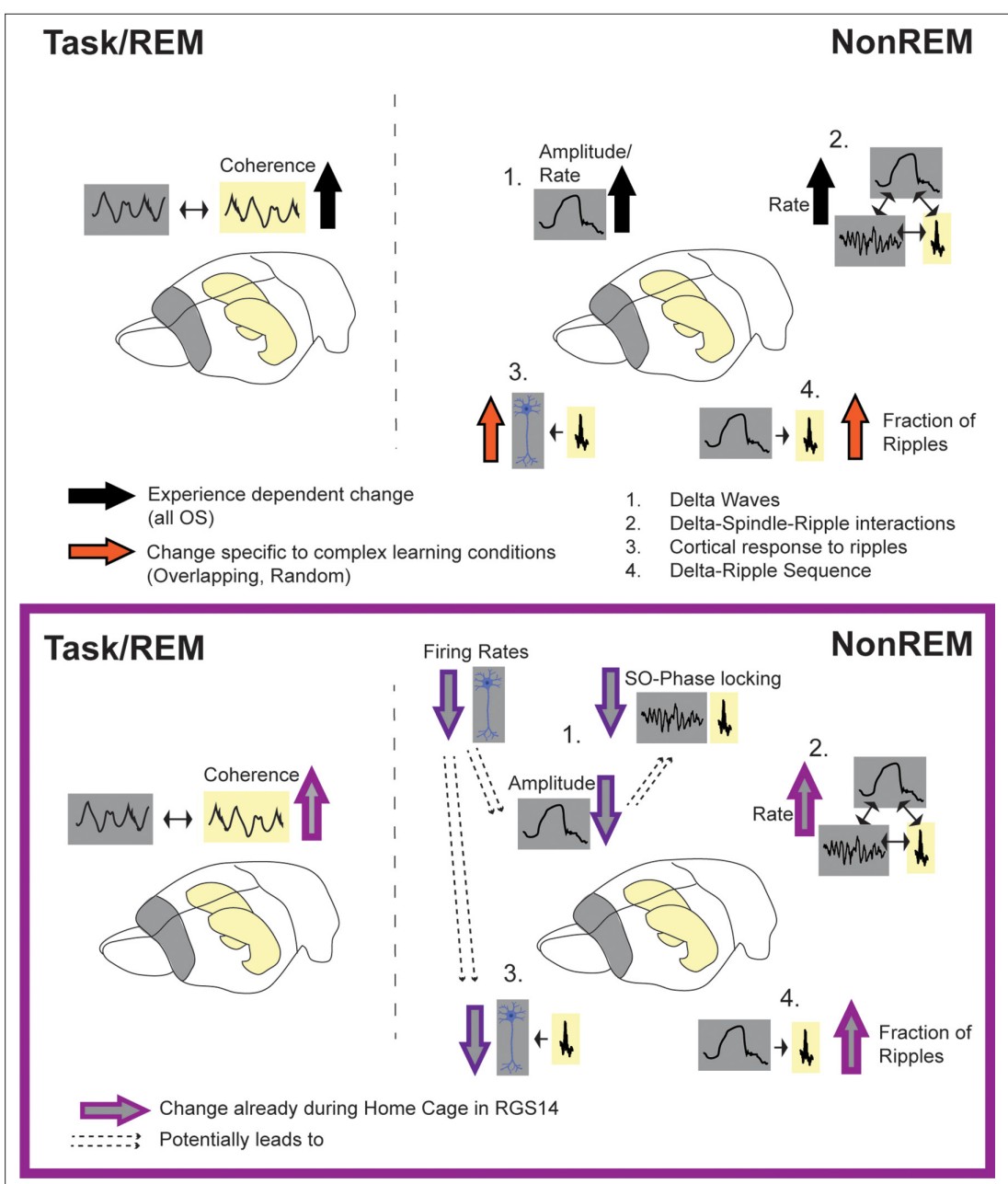

**Figure 6.** Summary of results. Top: effects of experience (black arrow, all OS vs HC) and complex learning (orange arrows, Overlapping/Random) in controls. Experience dependent effects: increased hippocampal-cortical theta-coherence during task and REM, increased delta-amplitude in NonREM (1) and increased spindle and delta rate leading also to more oscillatory coupling events (2). Complex Learning effects: increase number of neurons responsive to ripples (3) and increased fraction of ripples following a delta-wave (4). Bottom: Changes in RGS14. Learning effects already seen in Home Cage (light purple): increased theta coherence, increase oscillatory coupling rate (2) and increased Delta-Ripple fraction (4). But decreased, hippocampal-like firing rates lead to smaller cortical delta-waves (more similar to hippocampal delta-waves) and less slow oscillation (SO) phase locking (1) as well as decreased cortical response to ripples (3).

By contrasting these different conditions, we can extrapolate which effects are driven by which type of learning or experience. Changes induced by all OS conditions in comparison to Home Cage, should represent more general experience-dependent effects enabling simple memory consolidation or homeostasis (*Figure 6* black arrows). Here, these were (1) increased hippocampal-cortical theta-coherence during task and REM, (2) increased delta-amplitude in NonREM and (3) increased spindle and delta rate leading to more oscillatory coupling events. In contrast, changes seen specifically after Overlapping and Random should represent semantic-like memory consolidation with the comparison and integration of new (current trial) and old (previous trials) information (*Figure 6* Orange arrows). Here, these were (1) increased number of neurons responsive to ripples and (2) increased fraction of ripples following a delta-wave. Finally, contrasting Overlapping and Random would allow to isolate representations of the different statistical distributions. This would be applicable to for example decoding analysis including individual neural data. Regrettably, we did not record from enough neurons to allow for such an analysis here.

## Delta-spindle-ripple coupling

The difference between a general-experience driven increase in delta and spindle as well as most other oscillatory-coupling rates and the specific increase of ripples following delta waves only after complex learning, is especially interesting finding brought to light by the Object Space Task. Different types of interactions between delta-waves, spindles and ripples have been reported and emphasized by different labs. These are interactions between two oscillations such as delta followed by spindle (*Mölle et al., 2002*), delta followed by ripple (*Peyrache et al., 2009*), ripple followed by delta (*Maingret et al., 2016*), and spindles with a ripple in their troughs (*Sirota et al., 2003*), but also three-oscillation interactions such as delta followed by spindle with a ripple in the trough (*Diekelmann and Born, 2010*), delta followed by ripple then spindle (*Genzel et al., 2014*), ripple followed by delta and then spindle (*Maingret et al., 2016*). Until now, experiments tended to report only on one type of oscillation, making it difficult to compare results and create a framework for potential different or common functions of these different oscillation-couplings.

We had previously already observed that the increase of delta and spindles and their coupling was linked to general experience (also seen after novelty exposure) (*Aleman-Zapata et al., 2022*) and delta-spindle coupling is most often reported by researchers investigating human subject and applying simple memory paradigms (*Diekelmann and Born, 2010*; *Bastian et al., 2022*). This fits to our current results, where general coupling increased after our simple memory condition Stable. Of note, this general increase of delta and spindles also led to an increase of all delta-spindle-ripple coupling types. In contrast ripples following delta coupling increased specifically after the complex learning conditions. Previously this coupling was described together with prelimbic reactivations when animals learn complex rules (*Peyrache et al., 2009*). It would be tempting to speculate that delta-spindle and spindle with ripple coupling would facilitate simple learning while delta-ripple would correspond to consolidation of complex memories that rely on the comparison of new memories to other previously encoded experiences.

## RGS14-overexpression leads to learning after all experiences

Our hypothesis was that increasing cortical plasticity leads to "overlearning". Instead of distinguishing salient information and selectively consolidating, all experiences would be treated equal, consolidated and retained on the long-term in the cortex. Therefore, signatures of learning should already be present in control conditions such as Home Cage. We confirmed this hypothesis, both experience and complex-memory dependent learning effects observed in controls were already present in Home Cage in RGS14 (*Figure 6* light purple arrows). RGS14-overexpressing animals showed more hippocampal-cortical theta-coherence, increased oscillatory coupling rate and increased fraction of ripples following delta-waves. Thus, in RGS14 encoding and consolidation is already by default 'turned-on' instead of being experience or learning dependent. This overlearning would explain the increased interference effects seen behaviorally.

## RGS14-overexpression make the cortex hippocampal-like

Interestingly, while most signatures of learning were increased, some seemed to show more complex interactions (*Figure 6* dark purple arrows). In line with predictions by the SHY hypothesis (*Tononi*

*and Cirelli, 2014*; *Tononi and Cirelli, 2006*; *Vyazovskiy et al., 2009*) RGS14-overexpressing animals showed longer off-periods during NonREM sleep. However, contrary to predictions by the SHY hypothesis (*Tononi and Cirelli, 2014*; *Tononi and Cirelli, 2006*), delta waves were smaller in RGS14-overexpressing animals. The cause is likely the general decrease in firing rates accompanying increased plasticity. Slower-firing neurons were less phase locked to the slow oscillation and less neuronal firing in the upstate led to smaller delta waves (*Figure 2J*).

It has been previously proposed that the brain shows log-normal distributions and slow-firing neurons are more plastic than fast-firing ones (*Buzsáki and Mizuseki, 2014*). Slow-firing neurons, due to potentially weaker synapses, would be more sensitive to plastic changes (*Buzsáki and Mizuseki, 2014*). The current evidence, however, was mainly correlational and was only presented for the hippocampus (*Grosmark and Buzsáki, 2016*). Our results provide the first causal evidence that increasing synaptic plasticity in the cortex shifts the neural firing towards the slow firing end of the neural firing spectrum. Due to insufficient yield of the hippocampal tetrodes, we could not compare cortical to hippocampal firing rates directly in our recordings. However, others have shown faster firing rates in the entorhinal cortex in comparison to the hippocampus, and within the hippocampus faster firing rates in CA1 in comparison to CA3 and DG (*Buzsáki and Mizuseki, 2014*), potentially mirroring expected differences in plasticity. Interestingly, the change induced in our experiment – slowing of firing rates in RGS14 by one log magnitude – matches the firing rates reported for CA1 in comparison to entorhinal cortex in *Buzsáki and Mizuseki, 2014*. Thus, our firing-rates would indicate we did what we set out to do: turning the slow learner (cortex) into a fast learner (hippocampal-like neuronal firing). Interestingly, this was also seen in delta-amplitudes, where in RGS14 cortical delta-waves became as small as hippocampal delta-waves. Thus, our results highlight the complex interaction of proposed plasticity-related mechanisms present in firing-rates and sleep oscillations. It seems that slight changes in plasticity – as one would expect to occur during normal learning and experiences – lead to shifts as proposed by the SHY hypothesis. However, larger plasticity differences as expected for different brain areas or induced with artificial manipulations as applied here, lead to much larger shifts – in the case of delta-waves even in the opposite direction as natural learning – and complex interactions.

Interestingly, the change in firing-rate also influenced neuronal response to ripples. Overall ripple response was less in RGS14, despite ripple response per firing-rate group being similar in controls and RGS14. The slowest firing neurons generally did not show large ripple responses in both groups and RGS14 had more of these neurons, which is likely the reason for the overall decreased ripple-response.

## Top down control of ripples

While RGS14-overexpressing animals had more ripples and cortical-hippocampal interactions, the response of cortical neurons to ripples remained lower than in controls. The current prevalent view on hippocampal ripple oscillations is that they are essential for memory consolidation. Furthermore, it is believed that they are generated mainly as an outcome of the local synaptic interactions within the hippocampus inducing changes in several cortical areas and supporting the broadcasting of information from the hippocampus to the cortex (*Buzsáki, 2015*). Our results imply that there is an important top-down, cortical influence on this oscillation beyond the known local hippocampal mechanisms. Such a top-down influence has previously been shown for hippocampal theta sequences and ripples during task execution by inhibiting the prefrontal cortex with DREADDs (*Schmidt and Redish, 2021*; *Schmidt et al., 2019*).

Furthermore, and in alignment with existing results, it shows that hippocampus doesn't reactivate random incoming sensory information, instead it orients itself to the previous knowledge acquired by the cortex (*Rothschild et al., 2017*; *Wang and Ikemoto, 2016*). Fitting to this notion, we saw in controls that more hippocampal ripples followed cortical delta-oscillations after the Overlapping and Random condition in comparison to Stable or Home Cage. It has been proposed that the preceding delta-wave would carry the existing information that will then be updated by the incoming ripple information (*Genzel et al., 2014*). This would be needed in the Overlapping and Random condition to enable comparison of the current configuration to the previous configurations and thus the extraction of the statistical regularities. In RGS14 the control conditions (Home Cage and simple memory Stable) also showed this increase, further underlining the idea that in RGS14 all information is treated as salient independent of its relevance, which then would lead to increased interference effects.

## RGS14

Previous experiments using the plasticity-enhancer RGS14 in the peri-rhinal and visual cortex, showed increased conversion of short-term memory (40 min) to long-term memory (24 hr) (**Navarro-Lobato et al., 2021**; **Masmudi-Martín et al., 2019**). As in this study, treatment did not affect other behavioral parameters such as overall exploration times in a simple object exploration paradigm. They could show that overexpression of RGS14 led to increased BNDF at 60 min and 24 hr post training, which led to better long-term memory (**Navarro-Lobato et al., 2021**). The increase in BDNF in RGS14 resulted in more neurites and dendrite branching in pyramidal and non-pyramidal neurons (**Masmudi-Martín et al., 2019**). It would be tempting to speculate that the increased dendritic branching led to the decreases in firing rates. Increased branching with more synaptic connections would make it less likely for inputs to be synchronized and thus the neuron would be less likely to fire an action potential.

These previous experiments imply that the main effects of RGS14 on memory are the result of increased cellular plasticity and especially increased BDNF at encoding but also during consolidation. In the current results RGS14 overexpression had multiple effects in the brain, thus we cannot conclude if effects on behavior were directly due to the increase in cellular plasticity in the prelimbic cortex or indirectly induced by changing, for example, sleep-oscillations implicated in memory consolidation. This also is the case for the anisomycin treatment, which inhibits cellular consolidation via inhibition of protein synthesis. We did not record neural activity during the treatment; therefore, we cannot exclude that effects other than the inhibition of protein synthesis could negatively impacted consolidation after the interference trial and have contributed to the results. However, the main aim of the treatment was to impact the consolidation and thus retention of the interference memory, which was achieved.

Of note, with the current approach, we cannot disentangle if we see increased interference effects in RGS14 since they remember the interference-trial better than the original training and thus their behavior is more driven by the interference experience. Or if the effect is caused by the interference memory overwriting and thus eliminating the original training memory. Furthermore, the Granger analysis of directionality should be interpreted with caution since Granger can lead to spurious results especially in oscillating networks (**Kispersky et al., 2011**).

To conclude, we could show that increasing plasticity in the prelimbic cortex enhances the ability to retain one trial information but this negatively impacts abstracted, cumulative memory representations and the ability to distinguish learning experiences, confirming long-standing but unproven memory system theories. Furthermore, changes in cortical plasticity affect neuronal firing rates, hippocampal-cortical connectivity, cortical delta waves and hippocampal ripples.

# Materials and methods

## Study design

A total of 72 rats were used in this experiment: 64 in the behavioral experiment and 8 in the electrophysiological recordings. In each case animals were first extensively handled for multiple days (at least 3) until they did not flinch when the experimenter touched them (see handling videos on https://www.genzellab.com/). Next, all animals underwent viral-injection surgery (see below), half the animals received RGS14$_{414}$-lentivirus while the other had a vehicle (empty) lentivirus. In addition, 32 animals also received pharmacological canula's targeting prelimbic region during this surgery (16 vehicle, 16 RGS). These and the other 32 behavioral animals had a 2-week surgery recovery and then went on to do the habituation as well as training in the Object Space Task (all conditions counterbalanced within animal). The eight electrophysiology animals (four vehicle, four RGS) received a second surgery 3 weeks after the first one, for hyperdrive implantation. During 2–3 week surgery recovery, tetrodes were slowly lowered to target area before the animals also had habituation and training in the Object Space Task.

## Animals

Three-month-old male Lister Hooded rats weighing between 300 and 350 g at the experiment start (Charles Rivers, Germany) were used in this study. Rats were pair-housed in conventional eurostandard type IV cages (Techniplastic, UK) in a temperature-controlled (20+2 °C) room following a 12 hr light/dark cycle with water and food provided ad libitum. After lentiviral surgical intervention, animals

were single-housed for two days and paired with their cage mates after recovery in rat individually ventilated cages (IVC; Techniplastic, UK). Animals were maintained in their IVC in a barrier room for 14 days before downscaling them to conventional housing conditions. After hyperdrive implantation, rats were single-housed in until the end of the experiment. A total of 72 rats were used in this experiment: 64 in the behavioral experiment (n=16 per group, RGS14$_{414}$, vehicle, across two cohorts of 8–8 animals each; and n=16 per group RGS14$_{414}$-pharma, vehicle-pharma group across two cohorts of 8–8 animals each), and 8 in the electrophysiological recordings (n=4 per treatment, RGS14$_{414}$, and vehicle, split one animal per treatment across four cohorts). The behavioral experiments and electrophysiological recordings were performed during the light period (between 9:00-18:00).

All animal procedures were approved by the Central Commissie Dierproeven (CCD) and conducted according to the Experiments on Animals Act (protocol codes, 2016-014-020 and 2016-014-022).

## RGS14$_{414}$-lentivirus

The RGS14$_{414}$- and vehicle-lentivirus solutions (1.72x10$^7$ CFU/ml and 2.75x10$^6$ CFU/ml respectively) were prepared and provided by Dr. Zafaruddin Khan at the University of Malaga (Malaga, Spain) (*Masmudi-Martín et al., 2019*; *López-Aranda et al., 2009*; ). Briefly, the RGS14$_{414}$ gene (GenBank, AY987041) was cloned into the commercial vector p-LVX DsRed Monomer-C1 (Clontech, France) using DNA recombinant technology. Then, both non-replicant RGS14$_{414}$- and vehicle-lentivirus (empty vector) were prepared and titered using the Lenti-XTM (Clontech, France) according to the manufacturer's instructions.

The animal's procedures related to the non-replicant lentiviral solution were approved and carried out in compliance with institutional regulation.

## Tetrode hyperdrive

A customized lightweight tetrode micro-drive was manufactured to implant 10 and 6 movable tetrodes in the prelimbic cortex and hippocampus (HPC), respectively (*Brunetti et al., 2014*; *Kloosterman et al., 2009*; *Nguyen et al., 2009*). Two separate bundles of #33 polyimide tubes (Professional Plastics,) were prepared: one of 2 columns x 5 rows for prelimbic cortex and 3X3 for HPC. The bundles were fixed first to the customized 3D printed cannula and then into the customized 3D printed body drive. The 3D printed cannula was designed according to the Rat Brain Atlas in Stereotaxic Coordinates *Paxinos, 2013* for the correct placement of the bundles in the areas of interest. Inner tubes (#38 Polyimide tubes; Professional Plastics) were placed inside the outer tubes and glued to the shuttle, which moves through the body spokes thanks to an inox steel screw and a spring CBM011C 08E (Lee spring, Germany). A total of 16 tetrodes were built, twisting four 10 cm polyimide-insulated 12 µm Nickel-Chrome wires (80 turns forward and 40 turns reverse) (Kanthal Precision, Florida) and fused by heat. Tetrodes were loaded in the inner tubes, and their free ends were connected to a customized 64 channels, 24 mm round electrode interface board (EIB) using gold pins (Neuralynx). Previously, 2 NPD dual row 32 contact connectors (Omnetics) had been attached to the EIB. The tetrode tips were cut using fine sharp scissors (maximum length 3.5 mm and 3 mm for prelimbic cortex and HPC, respectively) and fixed to the inner tubes in the upper part. Tetrode tips were clean in distilled water and gold-plated (gold solution, Neuroalynx) using NanoZ software to lower their impedance to 100–200 kΩ and improve the signal-to-noise ratio. The tetrode tips were hidden at the same level as the bundle. The whole drive was covered with aluminum foil connected to the ground to reduce the electrostatic interference during the recordings. The bottom of the micro-drive was deepened in 70% ethanol for 12 hr before brain implantation.

## Stereotaxic surgeries
### Lentivirus injection

Lentiviral solutions were infused in the prelimbic cortex using stereotaxic surgery under biosafety level 2 conditions. The coordinates of the prelimbic cortex injection site were +3.2 mm AP,+/-0.8 mm ML from Bregma, and –2.5 mm DV from dura mater, according to The Rat Brain Atlas from Paxinos and Watson *Paxinos, 2013*. The procedure was carried out under isoflurane inhaled anesthesia. Unconsciousness was induced at 5% isoflurane +1 l/min O$_2$ and maintained at 1.5–2% isoflurane +1 l/min O$_2$. A 0.8 mm diameter craniotomy was drilled above the target area in each hemisphere. The DV dura mater coordinate was measured before performing the durotomy.

A 30 G dental carpule connected to a 10 µl Hamilton and an infusion pump (Micro-pump, WPI) was slowly inserted into the brain target area (0.2 mm/min). A total volume of 2 µl of the lentiviral solution was infused at 200 nl/min. After 5 min of diffusion, the needle was removed, and the incision was sutured.

For the anisomycin infusions, 32 animals were implanted with a 26 G bilateral guide cannula (3 mm length and 1.6 mm inter-cannula distance; Plastic1 Technology, USA) above prelimbic cortex (AP +3.2 mm, ML +/-0.8 mm from Bregma and DV –0.5 mm from dura mater *Paxinos, 2013*). Three supporting screws 1 mm X 3 mm were driven approximately 0.9–1 mm into the skull around the cannula. The bilateral cannula was slowly inserted into the brain target area (0.2 mm/min) after dura matter was removed. The whole structure was attached to the previously scratched skull by Metabond (Sun Medical, Japan) and simplex rapid dental cement (Kemdent, UK). The lentivirus infusions took place similarly as explained above, but using a 30 G bilateral internal cannula with a 2 mm projection length from the guide cannula (Plastic1 Technology, USA). The final DV coordinate was –2.5 mm from the dura mater. After 5 min of diffusion, the internal cannula was removed, the guide cannula was protected with a bilateral dummy cannula without projection and its cap.

Temperature, oxygen saturation, and blood pressure were monitored during the whole surgical procedure. Some eye cream (Opthosam) was applied to protect the corneas during the intervention. At the start and end of the surgery, 2 ml of 0.9% NaCl physiological serum was administered subcutaneously. As analgesia, animals were administered 0.07 mg/ml carprofen in their water bottles two days before and three days after surgery. Immediately before surgery, 5 mg/kg carprofen was sc injected. In addition, a mix of 4 mg/kg lidocaine and 1 mg/kg bupivacaine in a 0.9% NaCl physiological serum was administered sc locally in the head.

After the viral injection, animals were housed individually in rat IVC cages for 14 days. Their weights and status were monitored daily for the correct recovery of animals. Then, rats were pair housed with their previous cagemate and moved to conventional housing.

## Tetrode hyper-drive implantation

Twenty-one days after viral infusion, a second stereotaxic surgery took place for tetrode micro-drive implantation in 8 animals. The procedure was similar as described above. In this intervention additionally, a prophylactic 10 mg/kg sc injection of Baytril antibiotic was administered at the beginning of the surgery. Two craniotomies (2x1 mm and 1x1 mm for prelimbic cortex and HPC, respectively) were drilled above the target areas on the right hemisphere. The coordinates for the upper left corner of each craniotomy were: AP +4.5 mm and ML –0.5 (prelimbic cortex) and AP –3.8 mm and ML –2 mm (HPC) from Bregma (*Paxinos, 2013*). A ground screw (M1x3) was placed on the left hemisphere in the cerebellum (AP –11 mm, ML +2 mm from Bregma). In addition, six M1x3 mm supporting screws were driven and bound to the skull using Super-bond C&B dental cement (Sun Medical, Japan). Carefully, the durotomies were performed, and the brain's surface was exposed. Subsequently, the micro-drive was positioned on the brain's surface, and attached to the skull and the screws by simplex rapid dental cement (Kemdent, UK). Then, tetrodes were slowly screw-driven into the prelimbic area in prelimbic cortex (3 mm DV from brain surface) and the cortical layers above the HPC (1.5 mm DV from brain surface). The dorsal hippocampal CA1 pyramidal layer was reached progressively in the subsequent days.

## Pharmacological infusion

Anisomycin (ANI) powder (Merck, Germany) was solved in 1 M HCl in 0.9% NaCl physiological serum, and the pH was adjusted with 10 µl of 5 M NaOH. Aliquots were prepared and stored at –20 °C until the moment of use. Immediately after the 24 hr test in the object space task, animals were infused with 3 µl/hemisphere of a 25.6 µg/µl ANI solution or the solvent as control at 300 nl/min. An infusion pump and two 10 µl Hamilton syringes connected through a PE10 tube (Plastic1;) to a customized 30 G bilateral internal cannula with a 2 mm projection (Plastic1;) was used for the infusion. The internal cannula was carefully removed after 3 min of diffusion time, and the dummy and cap were placed back. All the animals from Experiment 2 received the infusion of both ANI and vehicle across different weeks for each experimental paradigm.

## Object Space Task

The Object Space Task (OST) is a newly developed behavioral paradigm to study simple and semantic-like memories in rodents (*Genzel et al., 2019*). The task is based on the tendency of rodents to

explore novel object-location in an open field across multiple trials. In these experiments, the OST took place as described previously (*Genzel et al., 2019*) at least 21 days after the viral infusion when the effect of the RGS14$_{414}$ protein is observed (*Masmudi-Martín et al., 2019*; *López-Aranda et al., 2009*; ). Briefly, animals were handled for 5 consecutive days before and after surgery recovery. Then, rats were accommodated in the experimental room and habituated to the open field across 5 sessions (one per day). In the first session, animals explored the open field with their cagemate for 30 min. In the rest of the session, each individual freely explored the open field for 10 min. Two Duplo objects were included in the open field center in the last two sessions to facilitate a better exploration time in the subsequent task.

The OST consists of two phases: a training phase of five training trials in which animals are exposed to two identical objects (different across trials) for 5 min (45–55 min intertrial time); and a interference/test phase consisting of a single 10 min' trial performed 24 hr and 72 hr after training. In the stable condition, both object locations were fixed during the training trials, and one object location was moved during the interference and tests sessions. In the overlapping condition, one object location was fixed, and the other one moved across training trials. In the interference and test sessions, the same object-location pattern from the last training trial was repeated.

The open field was a wooden square 75x75 x 60 cm. For the task, but not for the habituation, we placed 2D proximal cues on the open field walls and 3D distal cues above the open field. The cues were changed in each experimental session. In addition, the open field base colors changed across task sessions (white, blue, green, brown). The objects used vary in material (plastic, glass, wood, and metal), size, and colors. For electrophysiological recordings, not plastic objects were used to prevent static interference. All object bases were attached to 10 cm x 10 cm metal plates. Circular magnets were installed in the corners underneath the open field floor to prevent object movements during exploration. The open field and object surfaces were cleaned with 40% ethanol between trials to avoid odor biases.

Each trial was recorded using a camera above the open field. The object exploration time was manually scored *online* using the homemade software '*Scorer32*'. The experimenter was blinded for treatment and experimental conditions at the moment of scoring. Object locations and experimental conditions were counterbalanced across treatments, individuals, and sessions.

For electrophysiological recordings, we run four cohorts of 2 animals each. Each cohort included one vehicle- and one RGS14$_{414}$-treated animal, which performed the identical condition sequences with the same object-location patterns. Object locations and experimental conditions were counterbalanced across cohorts. Electrophysiological recordings took place during trials and rest periods (45 min before and 3 h after both training and test; and 45 min intertrial time during the training). Therefore, two brown wooden sleep boxes (40x75 x 60 cm height) with bedding material were placed next to the open field. Animals had been accustomed to the sleep box for at least 3 hr in each open-field habituation session.

Rats involved in the electrophysiological recordings also performed two experimental control conditions: homecage and random. The random condition was carried out as described previously (*Genzel et al., 2019*), so there was a lack of repetitive object location patterns across different trials. In the homecage, the animal was recorded for 7 hr and 10 min in the sleep box (a whole training session recording), and the experimenter kept the rat awake for the equivalent trial times.

## In vivo electrophysiology recordings

In vivo freely moving extracellular recordings were executed during the OST and the resting periods. One session per experimental condition (homecage, stable, overlapping, and random) was carried out per animal. The local field potential (LFP) and single-unit activity detected by the 64 channels were amplified, filtered, and digitized through two 32 channels chip amplifier headstages (InstanTechnology) connected through the Intan cables and a commutator into the Open Ephys acquisition box. The signal was visualized using the open-source Open Ephys GUI (sample rate 30 kHz). In addition, the headstage contains an accelerometer to record the movement of the animals.

## Tetrode electrolytic lesions

After all the recording sessions, the tetrode-implanted animals received brain electrolytic lesions 48 hr before the transcardial perfusion to identify the electrode tips placement. Thus, a current of 8 μA for

10 s was applied in two wires per tetrode using the stimulator with the animal under isoflurane inhaled anesthesia.

## Histology

### Brain processing

After data collection, animals had overdoses with 150 mg/kg sodium pentobarbital ip. Rats were transcardially perfused first with 80 ml of 0.1 M phosphate-buffered saline pH 7.4 (PBS) and then with 250 ml of 4% (w/v) paraformaldehyde in 0.1 M phosphate-buffered pH 7.4 (PFA). After brain extraction, it was immersed in PFA overnight at 4 °C. Then, the brains were rinsed in PBS 3 times for 10 min and cryoprotected by deepening in 20 ml of 30% (w/v) sucrose, 0.02% (w/v) $NaN_3$ in PBS. Once brains sank (after 2–3 days approx), they were frozen in dry ice and stored at –80 °C. Finally, 30 or 50 μm coronal sections of target areas were obtained using the cryostat (SLEE medical, Germany), collected in 48-well plates containing 0.02% (w/v) $NaN_3$ PBS and stored at 4 °C.

### Immunohistochemistry

The overexpression of $rgs14_{414}$ was checked by free-floating fluorescence immunohistochemistry. First, the target sections were selected, rinsed in PBS, and incubated overnight at 4 °C with the rabbit polyclonal anti-RGS14 antibody (Novus biological, NBP1-31174; dilution 1:500). Then, the Alexa fluor 488-conjugated goat anti-rabbit IgG (Life Technologies, A11008; dilution 1:1000) at room temperature for 2.5 hr. Some drops of water-soluble mounting medium containing DAPI (Abcam, ab104139) were applied for 5 min before placing the coverslip. Leica fluorescense microscope (Leica DM IRE2) and camera were used to observe and photograph the samples.

### Nissl staining

Coronal sections were AP sequentially mounted on gelatin-coated slides and incubated at 37 °C overnight. Slices were hydrated first in 0.1 M PBS pH 7.4 and then in Milli Q water for 20 min each. Next, brain sections were stained in 0.7 %(w/v) acetate cresyl violet for 20 min and dehydrated in an increasing ethanol gradient (water for 3 min, 70% ethanol for 20 s, 96% ethanol +acetic acid for 45 s, 100% ethanol for 5 min). Lastly, the tissue was immersed in xylene for 15 min, and the coverslip was placed using some DePeX mounting medium drops. Cannula placement, infusion traces, or/and tetrode lesions were observed and photographed under a light field microscope (Leica DM IRE2) and a camera.

## Behavioral data analysis

### Object Space Task

The total exploration time was calculated as the sum of the time spent exploring both object locations. The discrimination index (DI) was computed by subtracting the familiar object location exploration time to the novel object location and dividing it by the total exploration time. For Overlapping it is moved location – stable location, for Stable it is location-to-be-moved-at-test – stable location and for random which is assigned as moved and stable is random. A DI >0 means a preference for the new object location and consequently memory from the previous episode. A DI = 0 shows no preference for either the new object location or the fixed one. DI <0 means a preference for the stable object location.

### Model

The same computational model as in *Genzel et al., 2019* was used (see article for more detailed methods). In short, the model learns place-object associations and then translates this memory into an exploratory behavior: the objects that were stably found at the same location have a very low uncertainty and are thus either less attractive or more attractive (depending on the individuals) during exploration than objects found at changing locations (high uncertainty in place-object association). The source code of the computational model and model simulation/fitting procedures is available here: https://github.com/MehdiKhamassi/ObjectSpaceExplorationModel (copy archived at *Lobato, 2023a*).

The model employs two different parameters: a learning rate α, which determines the speed of memory accumulation; an inverse temperature β, which determines the strength and sign of memory expression during exploratory behavior.

A low learning rate α (i.e. close to 0) means that the model will need numerous repetitions of the same observation (i.e. in the Object Space Task, many trials observing the same place-object association) to properly memorize it. In contrast, a high learning rate α (i.e. close to 1) means that the model quickly memorizes new observations at the expense of old observations which are more quickly forgotten. As a consequence, with a low learning rate the exploratory behavior generated by the model will mostly reflect remote memories but not recent ones (semantic-like memory). Conversely, with a high learning rate, exploratory behavior in the model will mostly reflect recent memories but not remote ones (episodic-like memory).

Finally, an inverse temperature β close to zero means that the model does not strongly translate memories into object preferences for exploration, thus showing little object preference. In contrast, a high inverse temperature will mean that the model's exploratory behavior is strongly driven by differences in relative uncertainty between place-object associations. A high positive inverse temperature ($\beta$>0) will result in neophilic behavior: the model spends more time exploring objects associated with high uncertainty (i.e. novelty or constantly changing location); a high negative inverse temperature ($\beta$<0) will result in neophobic behavior: the model spends more time exploring objects with low uncertainty (stable/familiar objects).

The model was fitted to each mouse's trial-by-trial behavior using a maximum likelihood procedure described in *Genzel et al., 2019*, and similar to state-of-the-art model fitting methods in cognitive neuroscience (*Wilson and Collins, 2019*). In brief, this model fitting process found the best parameter values for each subject that best explain the relative proportion of time spent exploring each object at each trial. The main operations of the model are summarized in *Figure 1F*. All model equations are described in *Genzel et al., 2019*.

## Sleep architecture analysis

### Sleep scoring

Different states (wakefulness, NonREM, REM, and Intermediate) were *off-line* manually scored using, '*TheStateEditor*' developed by Dr. Andres Grosmark at Dr. Gyorgy Buzsaki lab. One channel per brain area (Prelimbic cortex and HPC) was selected per animal. Using a 10 s sliding window, an experienced researcher created the hypnogram and 1 s epoch vector indicating brain states. The absences of movement in the accelerometer discriminate between wakefulness and sleep. During sleep periods, a dominant theta frequency in the dorsal hippocampus in the absence of spindles and delta waves indicated REM sleep. NonREM sleep was classified when slow oscillations were detected in the prelimbic cortex. The intermediate phase was defined as short transitional periods between NonREM and REM that show an increase in frequency in the prelimbic cortex and frequency similar to theta in the dorsal hippocampus. Microarousals were defined later on as periods of wakefulness ≤15 s within a sleep period.

### Macroarchitecture

The total sleep time (TST), total wakefulness time, and total time for different sleep states (NonREM, REM, and Intermediate) were computed per session on MATLAB. The average across sessions was calculated per rat, and the mean and SEM were computed per treatment. Additionally, the % of TST of NonREM time, REM time, and intermediate time were calculated for the 3 hr recording post-training.

### Microarchitecture

The distribution of bout duration per stage was computed per treatment. The bouts number and duration were calculated per resting period and state. Only bouts longer than 4 s were considered for the analysis. The 1 s epoch state vectors of each session were concatenated with the interleaved trial wakefulness. NonREM episodes were defined as consecutive NonREM bouts without considering microarousals. Sleep period was described as an event of sleep between wake events >300 s. Each sleep period could include several NonREM periods defined as NonREM episodes without considering microarousals, or NonREM episodes followed by transitional stages and/or sleep cycles defined as NonREM episodes followed by transitional states and REM or NonREM followed by REM. In sleep,

periods can include quiet wakefulness <300 s. The number of sleep periods and the average duration per treatment were calculated on MATLAB. Additionally, the count and duration of NonREM periods and sleep cycles were also computed per treatment. The MATLAB scripts can be found at https://github.com/genzellab/sleep_architecture (copy archived at *Lobato, 2023b*).

## Local field potential analysis

### Signal preprocessing

For the following analyses, first a single channel was selected per brain area. For prelimbic cortex, the channel with the largest slow oscillations was chosen. For hippocampus, the channel closest to the pyramidal layer, which displayed noticeable ripples was selected. Both channels were originally acquired at a sampling rate of 30 kHz and to avoid working which such a high rate, the channels were filtered with a 3$^{rd}$ order Butterworth lowpass filter at 500 Hz to avoid signal aliasing and then downsampled to 1 kHz.

### Theta coherence

Theta coherence was computed as the magnitude squared coherence using the *mscohere function in* MATLAB and a custom-written script that collected the downsampled data from different animals and study days. The magnitude squared coherence was calculated as follows:

$$Cxy\left(T\right) = \frac{|Pxy(T)|}{Pxx(T)Pyy(T)}$$

where Cxy is the magnitude coherence, Pxy is the cross spectral density of the hippocampal and prelimbic cortex signal, Pxx is the hippocampal signal, and Pyy is the prelimbic cortexsignal. The coherence analysis focused on two periods of interest namely, WAKE and REM periods. Since REM sleeping periods might occur several times over several sleeping cycles, REM periods from pre and posttrial sleep were first extracted and concatenated together before running the analysis. For both Wake and REM periods, the power and cross-power spectra were computed on overlapping time windows of 1 s with 80% overlap. Then, the theta coherence was computed as the average value over the theta frequency range (5–12 Hz) for the Object Space (Stable, Overlapping and Random) and home cage conditions.

### Detection of spindles and delta waves

The downsampled prelimbic cortex channel (1 kHz) was loaded into the Matlab workspace and using a third-order Butterworth filter the signal was filtered to 9–20 Hz for detecting spindles and to 1–6 Hz for detecting delta waves. The NonREM bouts were then extracted from the filtered signal and concatenated. The functions FindSpindles and FindDeltaWaves from the Freely Moving Animal (FMA) toolbox http://fmatoolbox.sourceforge.net were modified and used to detect the start, peak and end of spindles and delta waves respectively. The optimal threshold was found for each animal by visually inspecting the detections and modifying the default parameters of the functions when needed. The results were saved as timestamps with respect to the concatenated NonREM signal in seconds. They were then used to find the timestamps with respect to the recorded signal. This process was repeated for pre and post trial sleep periods in study days pertaining to all animals in both treatment groups. The same method was applied for the detection of hippocampal delta waves.

### Ripple detection

The downsampled channels (1 kHz) of the hippocampal pyramidal layer were loaded into the Matlab workspace and the NonREM bouts were extracted. Using a third-order Butterworth bandpass filter, the epochs of HPC signal were filtered to a frequency range of 100–300 Hz. A custom MATLAB function was used for detecting the start, peak and end of the ripples by thresholding voltage peaks which lasted a minimum duration of 30ms above the threshold. The start and end of the ripple were determined as half the value of the selected threshold. Values equivalent to 5 times the standard deviation of concatenated NonREM bouts were computed individually for presleep and all post trials in a study day. The average of these values was calculated to find a single detection threshold per study day. An

offset of 5 microV was added to the threshold to reduce false positives. This was repeated for all study days pertaining to all animals in both treatment groups.

## Oscillations characteristics

The traces of each event detected (ripples, spindles, delta waves) were extracted using the start and end timestamps obtained from the detectors. The traces of the events were filtered in their corresponding detection frequency band. Characteristics such as the amplitude and mean frequency were calculated for these filtered events using built-in and custom MATLAB functions. Namely, the amplitude of the events was calculated by computing the envelope of the filtered trace using a Hilbert transform. The absolute value of the result was taken and its maximum was found. The mean frequency of the filtered traces was computed using the meanfreq function of MATLAB.

## Detection of oscillation sequences

The sequences between ripples, spindles and delta waves were counted in various combinations to study cortico-hippocampal coupling during NonREM sleep as done by *Maingret et al., 2016*. The time difference between the peaks of these events was compared to a fixed duration to establish if there was a sequential relationship in the following combinations of oscillations: Delta-Spindle (D-S), Delta-Ripple (D-R), Ripple-Delta (R-D), Ripple-Delta-Spindle (R-D-S), Delta-Ripple-Spindle (D-R-S). For D-S a sequence was considered when the duration between events was between 100–1300ms, for D-R it was 50–400ms and for R-D it was 50–250ms. To find R-D-S sequences, the results of R-D and D-S were compared to find delta waves preceded by a ripple and followed by a spindle. To find D-R-S sequences, the results of D-R and R-S (events between 2-1000ms) were compared to find ripples preceded by a delta wave and followed by a spindle. To find the Delta-Spindle with Ripple sequences, the results of D-S and spindles co-occurring with ripples (see next subsection) were matched to find spindles preceded by a delta and co-occurring with a ripple. The results were saved as counts of each sequence for each post-trial.

## Co-occurrence between ripples and spindles

The co-occurrence between ripples and spindles was computed by comparing the start and end timestamps of both events. To consider co-occurrence between a ripple and a spindle, either one of the following conditions had to be fulfilled. (1) A ripple had to start and end within the duration of the spindle. (2) One of the events had to start or end within the duration of the other. Given that more than one ripple can co-occur with the same spindle, we counted separately spindles co-occurring with spindles and spindles co-occurring with ripples.

## Slow oscillation phase

The downsampled prelimbic cortex signal was filtered in the 0.5–4 Hz range using a third-order Butterworth bandpass filter and its Hilbert transform was computed to find the phase angle of slow oscillations in a range from 0° to 360°. The peaks of ripples and spindles were then used to find the corresponding slow oscillation phase. This same signal was later used to find the phase during spikes timestamps of cortical neurons.

## Spectral analysis and Granger causality

A two-second-long window centered on each ripple peak was extracted from the hippocampus and the prelimbic cortex channels respectively. All ripples across animals and conditions were combined per treatment and their amplitude was computed by finding the maximum of their envelope computed with a Hilbert transform. The median ripple amplitude was calculated and the corresponding two-second-long windows of the 2000 ripples which amplitude was the closest to the median amplitude were included in the following analysis. A notch filter at 50 Hz was applied to the ripple-centered windows using the ft_preprocessing function from the Matlab-based Fieldtrip toolbox *Oostenveld et al., 2011*. The Short-time Fourier transformation was calculated to detect the changes of spectral power in hippocampus and prelimbic cortex with respect to a time window of ±1 s around each ripple. This was computed using the ft_freqanalysis function from Fieldtrip with a 100ms Hanning window and time steps of 10ms, for a frequency range from 100 to 300 Hz with a 2 Hz step for hippocampus and

from 0.5 to 20 Hz with a step of 0.5 Hz for prelimbic cortex. The resulting spectrograms were averaged and displayed. To statistically compare spectrograms between treatments, a nonparametric permutation test to correct for multiple comparisons with two-tailed pixel-based statistics was computed using 500 permutations and a p-value of 0.05 (**Maris and Oostenveld, 2007**).

To determine the predictive power between brain regions during ripples, the time-frequency Spectral Granger Causality was computed for each directionality (**Dhamala et al., 2008**). A window with length of 2.2 s centered around each ripple peak was extracted for the simultaneous hippocampal and prelimbic cortex signals. The length of this window was chosen to at least capture one cycle of 0.5 Hz activity. A two-second-long time-frequency non-parametric Spectral Granger causality was computed by implementing a Short- time Fourier transform with a 500ms Hanning window with 10ms steps using the Fieldtrip functions ft_freqanalysis and ft_connectivityanalysis respectively. To determine statistical differences between granger spectrograms, we created randomized trials by taking 400 random ripples per treatment and computing their time-frequency granger causality as described above. The result was stored, and the procedure was repeated 30 times to give a total of 30 randomized trials per treatment. We then used the trials of the rgs14 and control treatments to determine significant statistical difference in each pixel of the time-frequency matrix by applying a nonparametric permutation test to correct for multiple comparisons with a two-tailed pixel-based correction, using 500 permutations and a p-value of 0.05. The scripts used for LFP and the following neural activity analyses can be found at https://github.com/genzellab/rgs14, (**Lobato, 2023c** copy archived at swh:1:rev:c1b7c33cc10ac404a0d816a8e1ddeb3220edab58).

## Neuronal activity analysis

### Spike sorting

An important step of our analysis was identifying the cortical neurons and their spiking activity. The sleep and trial recordings of each tetrode of the prelimbic cortex were extracted and concatenated chronologically to allow tracking the individual neuronal activity across the whole study day. The tetrode recordings were preprocessed by applying a bandpass filter between 300 and 3000 Hz. For each tetrode, multiple spike sorting algorithms were run using the SpikeInterface python-based framework **Buccino et al., 2020**. The spike sorters used were Tridesclous, SpikingCircus, Klusta, HerdingSpikes, Ironclust, and MountainSort4. The default parameters included in SpikeInterface per spike sorter were used. Agreement between the spike sorters was computed and only putative neurons that were detected by at least two spike sorters were considered. When a tetrode didn't have any consensus, the detections of the MountainSort4 spike sorter were used given that this spike sorter had better scores according to the SpikeForest **Magland et al., 2020** metrics and our own examinations. After the putative neurons were detected, an experienced user curated manually the detections by visual inspection using the Phy interface and labeled them as either good, multiunit activity, or noise.

### Waveform extraction

Sampled at 30 kHz, the prelimbic cortex tetrode channels were loaded and then filtered using a third-order Butterworth bandpass filter with a frequency range of 300–600 Hz. Neurons labeled as 'good' were used for further analysis. A preliminary waveform per neuron was defined with an 82-sample window with 40 samples before and 41 samples after the spike timestamp (ST). For each neuron, STs were randomly permuted and a total of 2000 were selected. The permutation was done in order to avoid bias when selecting waveforms. These 2000 waveforms were averaged to obtain the mean spike waveform per neuron. Since tetrodes were used to record neuronal activity during the task and sleep, there were 4 average waveforms to choose from. The one with the highest peak amplitude was chosen. For each neuron, all STs were stored. Further, each neuron was assigned a unique ID. This process was performed for all neurons of all rats.

### Neuron classification

The average neuron waveforms calculated from rats 1, 2, 6, and 9 were categorized as 'Vehicle' and stored in a 139 (t1) ×82 matrix, with 139 being the total number of putative neurons and 82 being the number of samples as mentioned before. Similarly, 'RGS14' data from rats 3, 4, 7, 8 was stored as a 353 (t2) ×82 matrix. Variables t1 and t2 were the total counts of neurons for the respective treatments. The bz_CellClassification.m script from Buzsaki lab's GitHub was used to compute the

trough-to-peak delay time and the spike width for the spike waveform of each putative neuron. The function ClusterPointsBoundaryOutBW.m of the same repository was modified to incorporate visualization of interneuron and pyramidal data for both treatments. Neurons were then visualized in a 2D plane with x and y axes as trough-to-peak delay and spike width respectively. The set of coordinates of all putative neurons were fed into a Gaussian Mixture Model (GMM) with two components in order to find the centroids of the clusters of pyramidal cells and interneurons. The cluster which contained spikes with high values of spike width and trough-to-peak delay was labeled as the pyramidal neurons cluster, while the remaining one was labeled as the interneurons cluster. For both clusters, a threshold of mean +/- 2 SD with respect to their centroids was used to filter out the outliers. Next, the firing rates of the remaining neurons were calculated as is described in the following paragraph and those with extreme firing rate values were reinspected in the Phy interface by another experimenter to potentially discard remaining false positives. After this procedure there were a total of 101 pyramidal neurons for the RGS14 treatment and 57 for Vehicle. The total number of interneurons were 18 for RGS14 and 7 for Vehicle. In the following analyses, only pyramidal neurons were used given the low number of interneurons detected.

## Firing rate analysis

The spikes of each neuron were grouped by the sleep stage during which they fired. The total number of spikes of a neuron during a specific sleep stage were determined by counting all the spikes occurring during the sleep stage in question across pre-trial sleep, post-trial sleep and trials periods of a single day. The cumulative amount of time spent in a specific sleep stage during the day was determined similarly. Using these two values, the firing rate of a neuron during a sleep stage was computed by dividing the total number of spikes during the sleep stage by the cumulative amount of time spent in seconds in the sleep stage. The firing rate during each sleep and wake stage was calculated for all neurons. The firing rates of neurons during the 'Wake' stage in the vehicle control condition were divided into five quantiles based on their magnitudes (0–20%, 20–40%, 40–60%, 60–80%, 80–100%), as shown in *Figure 2G*. The upper-limit values of firing rate differentiating the groups in Vehicle were then used as a threshold to divide the neurons in the RGS treatments in five groups as well. The spike timestamps during NonREM sleep were collected across the whole study day for each neuron and the corresponding slow oscillation phases during each spike timestamps were extracted. The slow oscillation phase during spikes was calculated as described above for ripples and spindles.

## Detection of ON and OFF periods during NonREM sleep

The spiking activity of neurons was extracted for each NonREM bout of every pre and post-trial during a study day. The spike timestamps of the multiple neurons firing during a NonREM bout were combined to generate a single spike train of multiunit activity (MUA). A binary array with the length of the NonREM bout and a sampling rate of 30 kHz was created, where 1 meant a spike occurred during that timestamp and 0 meant absence of spike. The start and end of OFF periods were detected by calculating the inter-spike-interval (ISI) between spikes and finding values above 50ms. The remaining periods were considered for ON period detection. Only ON periods with a duration between 50 and 4000ms were kept. A minimum number of 10 spikes per ON period was required to include it in the analysis. Durations of all ON and OFF periods found during a pre or post-trial were computed and their mean value was stored. This was repeated for all study days in which pyramidal neurons were detected. Only ON and OFF periods of study days including a minimum of 7 neurons were included.

## Cortical activity during ripples

Using the spike timestamps during NonREM sleep, the cortical pyramidal neuron response to hippocampal ripples was computed in a 2 s window defined around the peak of each ripple. For each ripple the activity of all pyramidal neurons detected during that day was extracted. The spikes timestamps in the 2 s window were normalized to vary from –1 to 1 s, where 0 was the ripple peak. After concatenating the normalized timestamps across all ripple-centered windows per neuron, they were binned in 10ms bins and the number of spikes was determined for each bin. Hence, for each neuron, a [1x200] column vector was obtained. This vector was then z-normalized. The final z-scored vector was found by averaging the z-scored vector over all neurons. The data was smoothed twice using the MATLAB smooth function. To visualize the activity of prelimbic cortex pyramidal neurons around the ripple, the

firing activity of a neuron was determined from a 50ms window around the peak of the ripple (–20 to +30ms) by quantifying the number of spikes. After compiling the firing activity around the ripple peak for each neuron the values were sorted in an ascending order and was visualized using the MATLAB imagesc function.

## Figures

Figures were generated in GraphPad Prism (truncated violin plots with high smoothing).

## Acknowledgements

We thank Pelin Özsezer, Barbora Roldanus and Kopal Agarwal for data analysis, Mònica Siuraneta, Vera Mascarenhas Pombeiro Duarte Silva, Jay Chen, Alysha Maurmair, Jet van der Stoep, Joris van Hout, Koen van den Berg and Rian Kraan for performing behavioral experiments. Branco Weiss Fellowship – Society in Science (LG), VIDI NWO (LG), Fundación Alfonso Martín Escudero Fellowship (INL)

## Additional information

### Funding

| Funder | Grant reference number | Author |
|---|---|---|
| Fundacion Alfonso Martin Escudero Fellowship | | Irene Navarro Lobato |
| Branco Weiss Fellowship – Society in Science | | Lisa Genzel |
| Nederlandse Organisatie voor Wetenschappelijk Onderzoek | VIDI | Lisa Genzel |

The funders had no role in study design, data collection and interpretation, or the decision to submit the work for publication.

### Author contributions

Irene Navarro Lobato, Conceptualization, Data curation, Formal analysis, Supervision, Funding acquisition, Investigation, Visualization, Methodology, Writing – review and editing; Adrian Aleman-Zapata, Data curation, Formal analysis, Supervision, Investigation, Visualization, Writing – review and editing; Anumita Samanta, Investigation, Writing – review and editing; Milan Bogers, Investigation; Shekhar Narayanan, Formal analysis; Abdelrahman Rayan, Formal analysis, Methodology, Writing – review and editing; Alejandra Alonso, Jacqueline van der Meij, Supervision, Investigation; Mehdi Khamassi, Formal analysis, Visualization, Methodology, Writing – review and editing; Zafar U Khan, Resources, Methodology, Writing – review and editing; Lisa Genzel, Conceptualization, Formal analysis, Supervision, Funding acquisition, Visualization, Writing - original draft, Project administration, Writing – review and editing

### Author ORCIDs

Adrian Aleman-Zapata http://orcid.org/0000-0002-9894-4370
Shekhar Narayanan http://orcid.org/0000-0001-7609-2042
Abdelrahman Rayan http://orcid.org/0000-0002-0457-6379
Jacqueline van der Meij http://orcid.org/0000-0002-1083-8208
Lisa Genzel http://orcid.org/0000-0001-9537-7959

### Ethics

All animal procedures were approved by the Central Commissie Dierproeven (CCD) and conducted according to the Experiments on Animals Act (protocol codes, 2016-014-020 and 2016-014-022).

### Decision letter and Author response

Decision letter https://doi.org/10.7554/eLife.84911.sa1
Author response https://doi.org/10.7554/eLife.84911.sa2

# Additional files

## Supplementary files
• MDAR checklist

## Data availability
All data is available at https://osf.io/7xw43/. Code can be found on GitHub (see materials and methods for links).

The following dataset was generated:

| Author(s) | Year | Dataset title | Dataset URL | Database and Identifier |
|---|---|---|---|---|
| Navarro-Lobato I, Aleman-Zapata A, Genzel L | 2023 | RGS14_short | https://osf.io/7xw43/ | Open Science Framework, 7xw43 |

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
