## [Editor Report]

This important study reveals that slow plasticity in the neocortex is essential to prevent memory interference. The method of artificially increasing plasticity in the prefrontal cortex of rats during learning and its effect on sleep physiology, when memories are believed to be reprocessed, is solid. The work will be of interest to neuroscientists interested in learning and memory.

---

## [Decision Letter]

**Decision letter after peer review:**

Thank you for submitting your article "Learning Fast and Slow: Increased cortical plasticity leads to memory interference and enhanced hippocampal-cortical interactions" for consideration by *eLife*. Your article has been reviewed by 3 peer reviewers, and the evaluation has been overseen by a Reviewing Editor and Laura Colgin as the Senior Editor. The following individuals involved in the review of your submission have agreed to reveal their identity: Usman Farooq (Reviewer #2); Marcos Frank (Reviewer #3).

Essential revisions:

In their study, Navarro Lobato et al. explored the relationship between plasticity and memory in the hippocampo-cortical network. To this end, virally-expressed RGS was injected into the medial prefrontal cortex of rats, and animals were trained on behavioral assays to test different types of memories, namely semantic and episodic. The study reports that enhanced plasticity in the prefrontal cortex led to the interference of recent memories, providing evidence that too much plasticity – and consequently fast learning – was detrimental to the acquisition of semantic-like memories in the cortex. This increased plasticity was also associated with changes in sleep physiological patterns and hippocampal-cortical coupling.

While the reviewers agree that the study is of potential interest, they raised several major concerns, which are summarized below. They have agreed that the below concerns must be addressed to warrant the publication of the study.

1. It is important to clarify the behavioral results by providing a more detailed description of the task parameters (for example the type of objects used, etc.). Furthermore, the lack of discrimination in control animals is surprising and seems at odds with previously published results. Last, whether increased plasticity leads to increased interference and/or enhanced one-shot is unclear.

2. The study should provide more details on RGS over-expression, the exact site of injection, possible resulting anatomical changes (e.g. dendrites, at least a discussion on this point if no data are available), and justification that RGS does not lead to a ceiling effect in some task parameters (and not directly improve plasticity).

3. Electrophysiological results do not compare overlapping and stable conditions (while it seems these data were collected), these results should be presented or, at least, this choice of the presentation should be discussed.

4. The study should provide evidence that the reported decrease in firing rates does not depend on differences in behavior.

5. The relationship between plasticity and firing rates should be toned down, especially since the discussion is based on hippocampal data, for which the physiology may be different as in the cortex.

6. The study should provide a better characterization of changes during sleep, especially UP/DOWN state duration, REM sleep duration, and firing rates, as well as a comparison of PFC and hippocampus.

7. If possible, the study should include some physiological characterization of anisomycin-treated animals.

8. Potential alternative interpretations should be discussed, for example, the consequence of RGS over-expression on sleep architecture and physiology, which would itself change learning (not just a change in plasticity).

*Reviewer #1 (Recommendations for the authors):*

1) This is a difficult paper to read, as the authors present a mixture of results and discussion while describing figures in the text, with relatively little space to present the results. The presentation can be significantly improved to enhance readability by devoting the Results section to primarily describe results, and moving the interpretation/ speculation to the Discussion section as much as possible. In addition, it will be valuable to describe in detail the Object-Space task with a test trial after 24 hours as per the original experimental design (Genzel et al. 2019), before moving on to the interference condition. Finally, dividing multiple-graph panels into labelled sub-panels that are referenced in the Legend (eg., Figure 4) will be useful.

2) For the behavioral results in Figure 1C, the Overlapping condition is supposed to show slowly increasing DI corresponding to accumulated memory as per the text/ legend. Statistics need to be presented here, and it instead looks like there is an increase in DI for the Overlapping condition only on the 5th training trial. Further, the interference trial after 24 hours and the test trial after 48 hours should correspond to the test trial in the original design, where an increase in DI is expected, but none seems to be observed here. As noted in the Public review, the authors need to discuss this in detail.

3) Relatedly, please explain the colors and shades used in Figure 1D in detail, or change the color scheme. The dark gray alluded to here (Experiment 2 Control) looks like violet.

4) Line 196-198: "Furthermore, because it is the faster-firing neurons that dominate upstate spiking activity and therefore δ amplitude, the slowing of firing rates in the more plastic neurons is most likely the cause of the smaller δ waves seen in these animals". Can a reference be provided for the first part of the statement?

Also, reference 14 relates to hippocampal slow-firing neurons (as noted above), so any conclusions regarding slow-firing cortical neurons being more plastic need to be justified.

5) For physiological differences, is there any reason why data is not presented comparing the Stable and Overlapping OST condition since this is the central point of the manuscript? As per Methods on pg. 500-506, recordings were obtained separately for each condition.

6) The Granger analysis of directionality should be interpreted with caution since Granger can lead to spurious results (Kispersky et al. 2011), especially in oscillating networks.

7) In Figures S6 and S7, none of the axes labels and axes tick labels are visible – they are too small.

8) In Figure 4, there is no scale bar for the color plots. In Figure 4C, please explain the acronyms B, Pre, Post.

*Reviewer #2 (Recommendations for the authors):*

– Check the whole manuscript for grammar e.g., these theories 'provides'… they 'remains' (page 2, lines 46-49)

– Describe the object space task in more detail (specifically why the interference protocol was designed this way) in the main manuscript as it is crucial to the manuscript

– Which statistical tests were used to test for the hypothesis? Please provide details alongside each test (t-test statistics, sample size, etc.)

– How is this a semantic-like task? Please describe/explain in the manuscript.

– As a host of electrophysiological changes have been observed, a cartoon model might help show how they all fit together to support the hypothesis of the authors

– Are hippocampal firing rates different or only prelimbic?

– Why were the electrophysiological differences of the object space task not compared with the random condition but the home cage condition instead? To determine the neural signature of increased plasticity due to the task in the RGS14414 group wouldn't that comparison be best?

*Reviewer #3 (Recommendations for the authors):*

This is a very interesting and thoroughly performed study that addresses very important questions in the field. The authors use a viral-vector-based strategy to over-express a construct that increases dendritic arborization, and spine formation (in the cortex) and is known to boost certain forms of memory. This is interpreted as a means of increasing cortical plasticity. Then, through a combination of behavioral tasks, electrophysiology, and sleep analyses, they report numerous changes in cortical and hippocampal activity. There are a few areas of improvement that would greatly strengthen the study.

1. Can the authors address the mechanism governing the effect of RGS? It seems that (based on other studies) transfected brain areas show a general increase in dendritic arborization, spine increase, etc., but it's not clear how this translates to 'experience-dependent' plasticity per se. It seems like non-specific effects, akin to what happens when neurotrophins are injected into V1. The result of the latter is not always normal, or like what happens with naturally occurring plasticity. This may be occurring in this experiment as well, as RGS appears to generally increase hippocampal-cortical connectivity in a manner different than what is driven by the experience. Also, considering that a Lenti virus was used, which can lead to random genome integration, some discussion would be helpful.

2. Is there any histology showing the localization of the RGS transfection?

3. It seems the best control for the effects of RGS is to use a compound that targets the same pathways. For example, Navarro-Lobato recently used the knockdown of 14-3-3ζ gene for this purpose (JNS 2021). I ask because anisomycin probably involves other pathways (protein synthesis inhibition, apoptosis).

4. What happened to firing rates during REM sleep? Cortical neurons change their activity in REM sleep, possibly more so after 'plasticity'. These analyses would be very helpful.

5. Can the authors present data on the duration of NREM up and down states? I ask because this parameter has been measured previously (Vyazovskiy) in the context of sleep need and plasticity.

6. Is it possible that the lack of experience-dependent changes in the RGS animals (in some parameters) is due to a ceiling or masking effect? That RGS saturates the capacity for further change?

7. It would be helpful to indicate in the figure legends for the supplementary figures when the data are from RGS or controls.

8. What happened to EEGs, FRP's, etc., in the anisomycin-treated rats? This would help one determine if the cortex was abnormal in some way.

---

## [Author Response]

Essential revisions:In their study, Navarro Lobato et al. explored the relationship between plasticity and memory in the hippocampo-cortical network. To this end, virally-expressed RGS was injected into the medial prefrontal cortex of rats, and animals were trained on behavioral assays to test different types of memories, namely semantic and episodic. The study reports that enhanced plasticity in the prefrontal cortex led to the interference of recent memories, providing evidence that too much plasticity – and consequently fast learning – was detrimental to the acquisition of semantic-like memories in the cortex. This increased plasticity was also associated with changes in sleep physiological patterns and hippocampal-cortical coupling.While the reviewers agree that the study is of potential interest, they raised several major concerns, which are summarized below. They have agreed that the below concerns must be addressed to warrant the publication of the study.1. It is important to clarify the behavioral results by providing a more detailed description of the task parameters (for example the type of objects used, etc.). Furthermore, the lack of discrimination in control animals is surprising and seems at odds with previously published results. Last, whether increased plasticity leads to increased interference and/or enhanced one-shot is unclear.

We apologize that not all things were clear, we seem to have oversimplified some explanations. In the previous publication with the same task, a 24h test trial was used (not 72h after training as here). Actually, the interference trial in this project is behaviourally the same as the test trial in the previous publication. The reason we call it interference and not test, is simply because our main interest in the current project was how this trial affects the 72h test trial. However, we can analyze the behaviour in that trial to “replicate” our previous findings. This data was already included in the supplement (Figure S1) but we now expanded the figure to include a bar graph version that is similar to the Plos Bio figure of the original report. Here you can see that in vehicles we replicate previous findings and they are above chance, further there is not difference between the groups (that could have influenced our results at 72h). We also now include examples of objects used in this figure (see Author response image 1) and expanded the task description in the introduction and discussion. Finally, we agree that we cannot differentiate between increased interference/overwriting vs stronger behavioural control of the one-shot memory, which we now mention in the discussion.

**Author response image 1. sa2fig1:** 

2. The study should provide more details on RGS over-expression, the exact site of injection, possible resulting anatomical changes (e.g. dendrites, at least a discussion on this point if no data are available), and justification that RGS does not lead to a ceiling effect in some task parameters (and not directly improve plasticity).

The injection site was already included in the methods (The coordinates of the prelimbic cortex injection site were +3.2 mm AP, +/-0.8 mm ML from Bregma, and -2.5 mm DV from dura mater) and now we added a histology overview. We do not have data on anatomical changes but now include a paragraph in the discussion citing previous data and findings. We expanded the discussion on how RGS on one hand shows “overlearning” but on the other hand more complex interactions of some learning mechanisms. Further, we include discussion on other parameters such as overall exploration time and how RGS14 does not change these.

3. Electrophysiological results do not compare overlapping and stable conditions (while it seems these data were collected), these results should be presented or, at least, this choice of the presentation should be discussed.

Our starting hypothesis was that RGS14 animals show “overlearning”, so every experience is treated as a salient learning experience. Therefore, we expected the biggest difference between treatments for the home cage condition (HC) and thus focusses on the HC vs OS analysis. Since as predicted this analysis provided the main effects and a large volume of results, we did not delve in more depth in the conditions.

However, in the meantime in a different project with the same task (including conditions HC, overlapping and stable but no RGS treatment or random training condition) we have explored the condition effect in more detail and found a specific effect of the percentage of ripples that occur after a δ wave in the Overlapping training condition. Since the reviewers pointed out this shortcoming, we now applied the same analysis to the RGS data-set and are happy to report that we find the same effect (included in the revision). We further now split the ripple-response analysis for condition and can also report differences there. Therefore, we would like to thank the reviewers for encouraging us to continue with the analysis of the data in more detail, considering new findings in our other projects. We have added a new figure and expand on this in the discussion.

4. The study should provide evidence that the reported decrease in firing rates does not depend on differences in behavior.

Firing rate were shown for different wake and sleep states (normalized to task wake) in the supplement. We now added firing rate group analysis for wake in the sleep box (no specific behaviour), NREM and REM (Figure 1S1 and2). All analysis confirms the initial finding from task wake, therefore differences in firing-rates are not due to differences in behaviour.

5. The relationship between plasticity and firing rates should be toned down, especially since the discussion is based on hippocampal data, for which the physiology may be different as in the cortex.

We adapted the discussion on firing rates and now include also more referencing to the original review proposing the idea that goes beyond hippocampus. We also now point out that the previously cited paper is focussed on the hippocampus. Overall, we expanded the discussion including also the new results in regards to hippocampal δ-waves and reported hippocampal-firing-rates (see answer to next comment).

6. The study should provide a better characterization of changes during sleep, especially UP/DOWN state duration, REM sleep duration, and firing rates, as well as a comparison of PFC and hippocampus.

We added REM bout duration (overall REM sleep duration was already included in the original figure 1) and are happy to report that there is a significant difference. We had originally not followed up on bout duration since there was no effect in overall duration. Regrettably we cannot add a PFC and HPC comparison for the firing rates, since we did not have sufficient tetrodes correctly placed in the hippocampus to record enough neurons to make this analysis reliable.

However, we now expand the discussion on the hippocampal and cortical firing rates and now mention that the magnitude of change we see in RGS14 would mirror the firing rate changes we would expect for the hippocampus according to the literature.

A HPC-PFC comparison we could add, was the analysis of δ-waves. Interestingly, in controls hippocampal δ waves are smaller than cortical ones. Further, the decrease in amplitude we already reported for RGS14, cortical δ-waves, corresponds to the normal size of hippocampal δ-waves (see new Figure 2). Thus, in RGS14 cortical and hippocampal firing-rates and δ-amplitudes are the same, which is not the case for controls. This is further evidence that with our manipulation we turned our “slow learning” cortex into a plastic, hippocampal-like structure.

Finally, we also added the on/off period analysis. Interestingly, in RGS14 off periods are longer, fitting to the SHY hypothesis. Further, we can replicate the observed time course of on/off durations reported previously (included in new Figure 2).

7. If possible, the study should include some physiological characterization of anisomycin-treated animals.

Regrettably, it was not possible to include electrophysiology in the anysomycin treated animals. We currently do not have implants that combine tetrodes with canula. We now include in the discussion a comment in that regard.

8. Potential alternative interpretations should be discussed, for example, the consequence of RGS over-expression on sleep architecture and physiology, which would itself change learning (not just a change in plasticity).

We now included this in the discussion, mentioning that we cannot disentangle if our results are due to direct changes in cellular plasticity or via changes in e.g. oscillatory coupling or sleep architecture.

Reviewer #1 (Recommendations for the authors):1) This is a difficult paper to read, as the authors present a mixture of results and discussion while describing figures in the text, with relatively little space to present the results. The presentation can be significantly improved to enhance readability by devoting the Results section to primarily describe results, and moving the interpretation/ speculation to the Discussion section as much as possible. In addition, it will be valuable to describe in detail the Object-Space task with a test trial after 24 hours as per the original experimental design (Genzel et al. 2019), before moving on to the interference condition. Finally, dividing multiple-graph panels into labelled sub-panels that are referenced in the Legend (eg., Figure 4) will be useful.

We now expanded the OS task description (introduction/discussion), include labelled sub-panels and further moved interpretation/speculation out of results to improve readability. We still kept those sections that explain why we decided to do a certain analysis to help the reader understand our approach. Finally, we significantly expanded the discussion, hopefully improving clarity and including more structure.

2) For the behavioral results in Figure 1C, the Overlapping condition is supposed to show slowly increasing DI corresponding to accumulated memory as per the text/ legend. Statistics need to be presented here, and it instead looks like there is an increase in DI for the Overlapping condition only on the 5th training trial. Further, the interference trial after 24 hours and the test trial after 48 hours should correspond to the test trial in the original design, where an increase in DI is expected, but none seems to be observed here. As noted in the Public review, the authors need to discuss this in detail.

As mentioned above, we now clarified that the 24h interference is similar to 24h test of past experiments and that we replicate effects. The same is for training DI that over many projects tends to show the same curve (most publications to come soon), where in trial 2-4 there is only a slight increase and the main increase in in trial 5 when the newest location is shown (not shown in trial 1-4). We now included stats (significant condition and condXtrial interaction but no effect or interaction with treatment) and in Author response image 2 the same training figure without Stable, it makes it a bit easier to see that there is an increase starting trial 2.

3) Relatedly, please explain the colors and shades used in Figure 1D in detail, or change the color scheme. The dark gray alluded to here (Experiment 2 Control) looks like violet.

We now moved the color scheme description higher up in the legend, directly after mentioning the groups of 1D. It may have one some screens the impression of violet but it is programmed as dark grey.

4) Line 196-198: "Furthermore, because it is the faster-firing neurons that dominate upstate spiking activity and therefore δ amplitude, the slowing of firing rates in the more plastic neurons is most likely the cause of the smaller δ waves seen in these animals". Can a reference be provided for the first part of the statement?

We added to which results this was referring to (Figure 2J).

Also, reference 14 relates to hippocampal slow-firing neurons (as noted above), so any conclusions regarding slow-firing cortical neurons being more plastic need to be justified.

We expanded this section including a new citation about cortical neurons.

5) For physiological differences, is there any reason why data is not presented comparing the Stable and Overlapping OST condition since this is the central point of the manuscript? As per Methods on pg. 500-506, recordings were obtained separately for each condition.

This data and analysis has now been added as mentioned in the main comments.

6) The Granger analysis of directionality should be interpreted with caution since Granger can lead to spurious results (Kispersky et al. 2011), especially in oscillating networks.

We added this statement and citation to the discussion

7) In Figures S6 and S7, none of the axes labels and axes tick labels are visible – they are too small.

We remade these figures with larger fonts etc.

8) In Figure 4, there is no scale bar for the color plots. In Figure 4C, please explain the acronyms B, Pre, Post.

We added color scale bars. The acronyms were explained in the legend: for each types the response at baseline (200-120ms before ripple), pre-ripple (120ms-40ms before ripple), during ripple (40ms before – 40ms after ripple peak), and post-ripple (40ms^-1^20ms after ripple)

Reviewer #2 (Recommendations for the authors):– Check the whole manuscript for grammar e.g., these theories 'provides'… they 'remains' (page 2, lines 46-49)

We performed a language check on the manuscript and hope to have caught all mistakes.

– Describe the object space task in more detail (specifically why the interference protocol was designed this way) in the main manuscript as it is crucial to the manuscript

We expanded the description.

– Which statistical tests were used to test for the hypothesis? Please provide details alongside each test (t-test statistics, sample size, etc.)

These are provided in the figure legends (which test is used, sample size in the degrees of freedom etc) to facilitate readability of the main text.

– How is this a semantic-like task? Please describe/explain in the manuscript.

We now added this to the introduction and discussion.

– As a host of electrophysiological changes have been observed, a cartoon model might help show how they all fit together to support the hypothesis of the authors

We added a cartoon model (see Figure 6) and also now expanded the discussion with subheadings. We hope that the new subdivision of effects also helps focus the article. We would like to thank the reviewer for this suggestion, since the model made us aware which of our results from the supplement should rather be included in the main figures. We moved the ripple-response analysis per firing-rate group to Figure 4.

– Are hippocampal firing rates different or only prelimbic?

Regrettably, we did not manage to place enough tetrodes in the hippocampus to make such an analysis feasible. However, as shown in the main comments, this effect was previously reported by others. Our manipulation made the cortex have similar firing-rates as usually seen in the hippocampus.

– Why were the electrophysiological differences of the object space task not compared with the random condition but the home cage condition instead? To determine the neural signature of increased plasticity due to the task in the RGS14414 group wouldn't that comparison be best?

As can be seen in the new analysis, random actually behaves like overlapping. It makes sense since in both cases objects are moving and potentially a statistical pattern can be detected. This is also seen in similar learning rates for both conditions in the computational model in the original Plos Biology article. Stable is less of a learning incentive, since less novelty is present during the task. But they are still experiencing something and are still learning a simple task. We now expanded this in the discussion with an explanation of the different possible contrasts with these conditions.

Reviewer #3 (Recommendations for the authors):This is a very interesting and thoroughly performed study that addresses very important questions in the field. The authors use a viral-vector-based strategy to over-express a construct that increases dendritic arborization, and spine formation (in the cortex) and is known to boost certain forms of memory. This is interpreted as a means of increasing cortical plasticity. Then, through a combination of behavioral tasks, electrophysiology, and sleep analyses, they report numerous changes in cortical and hippocampal activity. There are a few areas of improvement that would greatly strengthen the study.1. Can the authors address the mechanism governing the effect of RGS? It seems that (based on other studies) transfected brain areas show a general increase in dendritic arborization, spine increase, etc., but it's not clear how this translates to 'experience-dependent' plasticity per se. It seems like non-specific effects, akin to what happens when neurotrophins are injected into V1. The result of the latter is not always normal, or like what happens with naturally occurring plasticity. This may be occurring in this experiment as well, as RGS appears to generally increase hippocampal-cortical connectivity in a manner different than what is driven by the experience. Also, considering that a Lenti virus was used, which can lead to random genome integration, some discussion would be helpful.

Yes! This is exactly the point we are trying to make. Naturally occurring plasticity would be selective for experience and using general “plasticity enhancers” would break down the natural regulation and processes. We tried to emphasize this more in the discussion by showing where RGS seems to be learning in “overdrive” and where effects break down due to the sledge-hammer approach (dark purple in the new comic summary).

2. Is there any histology showing the localization of the RGS transfection?

We now included an overview picture with injection site in the supplement (Figure 2S3) to complement the transfection picture in Figure 1. There is a lot of background in the immune signal, which is why positive transfection can only be confirmed in zoomed in pictures.

3. It seems the best control for the effects of RGS is to use a compound that targets the same pathways. For example, Navarro-Lobato recently used the knockdown of 14-3-3ζ gene for this purpose (JNS 2021). I ask because anisomycin probably involves other pathways (protein synthesis inhibition, apoptosis).

With the manipulation we aimed at generally inhibiting consolidation of the interference trial and not subtly disentangle how RGS14 works. We use RGS14 simply as a tool to manipulate plasticity but are not specifically interested how it does this (we leave this to the experts in Malaga). This is why we chose to use a “consolidation inhibitor” as is used most often in the memory field, anisomycin.

4. What happened to firing rates during REM sleep? Cortical neurons change their activity in REM sleep, possibly more so after 'plasticity'. These analyses would be very helpful.

This was added to the supplement, the same effects were seen. Already previously in the supplement we presented how each neuron group modulated their firing rate over the different states.

5. Can the authors present data on the duration of NREM up and down states? I ask because this parameter has been measured previously (Vyazovskiy) in the context of sleep need and plasticity.

We would like to thank the reviewer for this suggestion. We added the analysis and as SHY predicts we see longer Off-periods in RGS14 and we can replicate the time-course finding where off-periods become shorter over long sleep periods. We added the results to Figure 2 (see above).

6. Is it possible that the lack of experience-dependent changes in the RGS animals (in some parameters) is due to a ceiling or masking effect? That RGS saturates the capacity for further change?

We now added to the discussion that we cannot disentangle some of the effects. On one side the cellular consolidation vs sleep consolidation but then also increase interference memory vs actually overwriting the old memory (see also previous answers to comments of other reviewers). Overall, behaviour in the task was similar (exploration times etc) as was previously reported. We now include this also in the discussion.

7. It would be helpful to indicate in the figure legends for the supplementary figures when the data are from RGS or controls.

We added this where it was missing and apologize for the oversight.

8. What happened to EEGs, FRP's, etc., in the anisomycin-treated rats? This would help one determine if the cortex was abnormal in some way.

Regrettably we do not have combined anysomycin – electrophysiolgical data. We currently do not have an implant that can combine cannulas and recordings. However, we now add this to the discussion and also that our general aim was to inhibit consolidation.